# Mineralogical, Geochemical Characterization and Physicochemical Properties of Kaolinitic Clays of the Eastern Part of the Douala Sub-Basin, Cameroon, Central Africa

Kankao Oumla Oumar [1,*], Ngon Ngon Gilbert François [1,2], Mbog Michel Bertrand [3], Tehna Nathanael [4], Bayiga Elie Constantin [1], Mbaï Joel Simon [1] and Etame Jacques [1]

[1] Department of Earth Sciences, Faculty of Science, University of Douala, Douala P.O. Box 24157, Cameroon
[2] School of Geology and Mining Engineering, University of Ngaoundere, Ngaoundéré P.O. Box 454, Cameroon
[3] Department of Earth Sciences, Faculty of Science, University of Dschang, Dschang P.O. Box 067, Cameroon
[4] Department of Earth Sciences, Faculty of Science, University of Yaoundé I, Yaoundé P.O. Box 812, Cameroon
[*] Correspondence: kankaooumar@gmail.com

**Abstract:** The sedimentary clays of the Douala sub-basin (Cameroon) were studied to determine their mineralogical composition and physicochemical properties to boost their potential suitability as materials for traditional ceramics and eventually modern ceramics. These clayey materials are not widely used locally as building materials and little data are available on these materials in the field of ceramics and they are relatively unknown. Three profiles from 3.9 to 7.4 m thickness were studied on the field in order to determine their mineralogical (X-ray diffraction, infrared), chemical (X-ray fluorescence) and physicochemical (particle size, Atterberg limits, organic matter, cation exchange capacity and hydrogen potential) properties. Globally, ten (10) clay samples were analyzed to highlight the nature and technological properties of these clays. Mineralogically, kaolinite (48.3–69.2 wt.%) and quartz (20.5–41.2 wt.%) were the most abundant minerals in these raw clay materials. They were associated with a very small or moderate quantity of illite, hematite, goethite, feldspar, gibbsite and micas. Geochemically, the clayey materials had high silica ($SiO_2$, 22.21–58.03%) and alumina ($Al_2O_3$, 12.84–22.94%) contents, with a significant amount of iron oxides ($Fe_2O_3$, 1.07–17.92%). Other oxides ($K_2O$, $MgO$, $TiO_2$, $Na_2O$, $MnO$, $CaO$ and $P_2O_5$) were in a relatively lower proportion. A high level of alumina content explains the kaolinitic nature of these clayey materials. The results of the granulometric analysis of the clayey materials showed the following distribution: clay (26–99%) followed by silt (1–70%) and sand (0–4%). This corresponds to silty clay soils according to the Belgian textural classification diagram, with high plastic index (63.9%) characteristics. The studied clay materials are good candidates for the production of ceramics and terracotta building. This study is therefore important for any application of this type of clay in various industrial fields.

**Keywords:** Cameroon; clay material; Douala sub-basin; mineralogical; physicochemical

## 1. Introduction

The traditional use of clayey materials existed in ancient times in Africa (Egypt), Asia (China), America (Mexico), Europe (Roman), etc. [1]. Clay is a material which has a particle size less than 2 μm and belongs to the family of minerals with similar chemical compositions and common crystal structural characteristics [2]. Raw clay materials are mineral resources with major importance in industries because of their numerous uses [3,4]. Presently, more than a third of the world's population uses clayey products, due to their quality, weather resistance, plasticity and malleability. The behavior of clay materials is related to their mineralogy and chemical composition, associated with certain geotechnical characteristics (particle size, plasticity, etc.). Their chemical compositions vary depending on both the physical and chemical changes in the environment where clay deposits are found [5]. The use of raw clay materials has grown in several and varied applications

(paints, plastics, cosmetics, pharmaceuticals and the production of ceramic materials of wide distribution) [6–8]. Indeed, the fields of application of clay minerals are more and more numerous because they are used in several industrial sectors, the most developed of which are the ceramic industry for the manufacturing of porcelain and earthenware, the agricultural industry to dilute pesticides and the pharmaceutical industry for the manufacturing of medication [6–8]. Studies have been carried out on the applications of ceramic clays in several areas around the world [9–12]. However, several researchers have been interested in the ceramic applications of clay materials in Africa [13–15]. Several works have focused on raw clay materials in Cameroon. Clay materials have been identified in several geological environments in Cameroon [16–20]. Other works have been carried out on physicochemical and mineralogical characteristics of the catalytic properties and thermal behavior of some clay materials [21,22]. In Cameroon's coastal basins, studies were carried out for the physicochemical and mineralogical characterization of clay materials in order to understand their origin and paleoenvironment [23,24] and to determine their properties for various applications [25–27]. In the present context, the natural, technological and mineralogical properties of the clays in the Douala sub-basin are not very well know and few works have been conducted on the ceramic aspect.

The aim of this study is to carry out a mineralogical and physicochemical characterization of the clay materials at the eastern part of the Douala sub-basin (Cameroon, Central Africa) in order to determine their nature and identify their technological classification and suitability for various ceramic applications.

## 2. Geographical and Geological Setting

The study area is situated at the eastern part of the Douala sub-basin between 3°10′–4°12′ N and 9°50′–9°52′ E, which forms the northern part of the Cameroonian Douala/Kribi-Campo basin in the Gulf of Guinea (Figure 1). It is limited to the north by the Cameroon Volcanic Line (CVL), to the south by the Rio Muni Basin (RMB) (Equatorial Guinea) and to the east by the Cameroon Pan-African basement [28]. The tectono-stratigraphic history of the Douala sub-basin includes several episodes of tectonic activity, which can be grouped in two great stages: (1) in the first stage (Early Cretaceous–upper Oligocene), basin development and sedimentation were related primarily to rift-associated tectonics. This first stage can be divided in two phases (late Aptian–Turonian and Senonian–Eocene) and two erosive phases (Senonian unconformity and mid-Eocene to base Miocene); (2) in the second stage (upper Oligocene–Recent), basin development and sedimentation were related primarily to the development of the Cameroon Volcanic Line, particularly to the development of the Oligocene to Recent volcanoes [29,30]. The study area is situated in the N'Kapa formation (Paleocene–Eocene), Matanda Formation (Miocene) and Wouri Formation (Plio–Pleistocene), which are constituted of marl, clays, sands and fine sandstone [30,31].

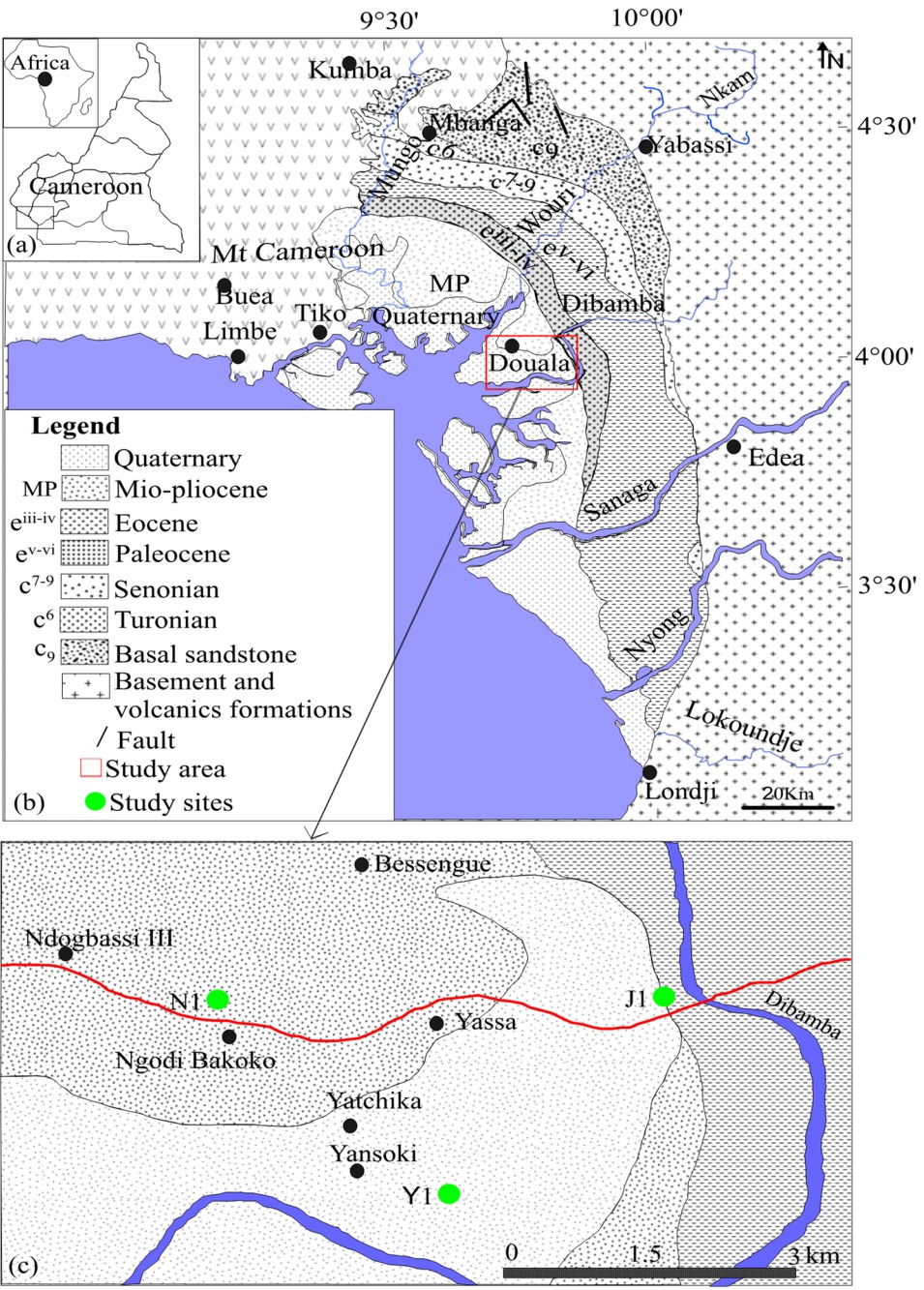

**Figure 1.** Geological representation of Douala modified [30]: (**a**) geolocation map of the Douala sub-basin, Littoral Cameroon; (**b**) geological representation of the Douala sub-basin containing the different study sites; (**c**) geological representation of the study area.

## 3. Material and Methods

### 3.1. Samples and Sampling Process

Three holes (Y1, J1 and N1) measuring 8.7, 8.8 and 6.5 m deep, respectively, were dug with a spade according to the availability of the study sites and the positions at which some craftsmen exploit clay materials for their activities. Each profile was described and divided based on their texture and color (Munsell color chart) in different horizons or layers in the vertical sections. These profiles are essentially constituted of clayey horizons situated on the bottom of the profiles. All the profiles are surmounted by some sandy or conglomerated horizons, which may alternate with the rusty horizons and organo-mineral horizon. After describing the profiles, seventeen (17) samples of 5 kg each were taken manually with a

shovel in the center of the different horizons encountered (Figure 2). The choice of the ten (10) samples for the analyses were essentially focused on the clayey levels. Thus, ten (10) representative samples from the study area were grouped into five facies (Figure 3): the dark gray facies (Y1C1 and N1C1); light gray facies (J1C1 and N1C2); purplish facies (J1C2); the yellow facies (Y1C2 and N1C4); and the multicolored facies (Y1C3, J1C3 and N1C3). Each representative sample was subjected to a mineralogical, geochemical and physicochemical analysis.

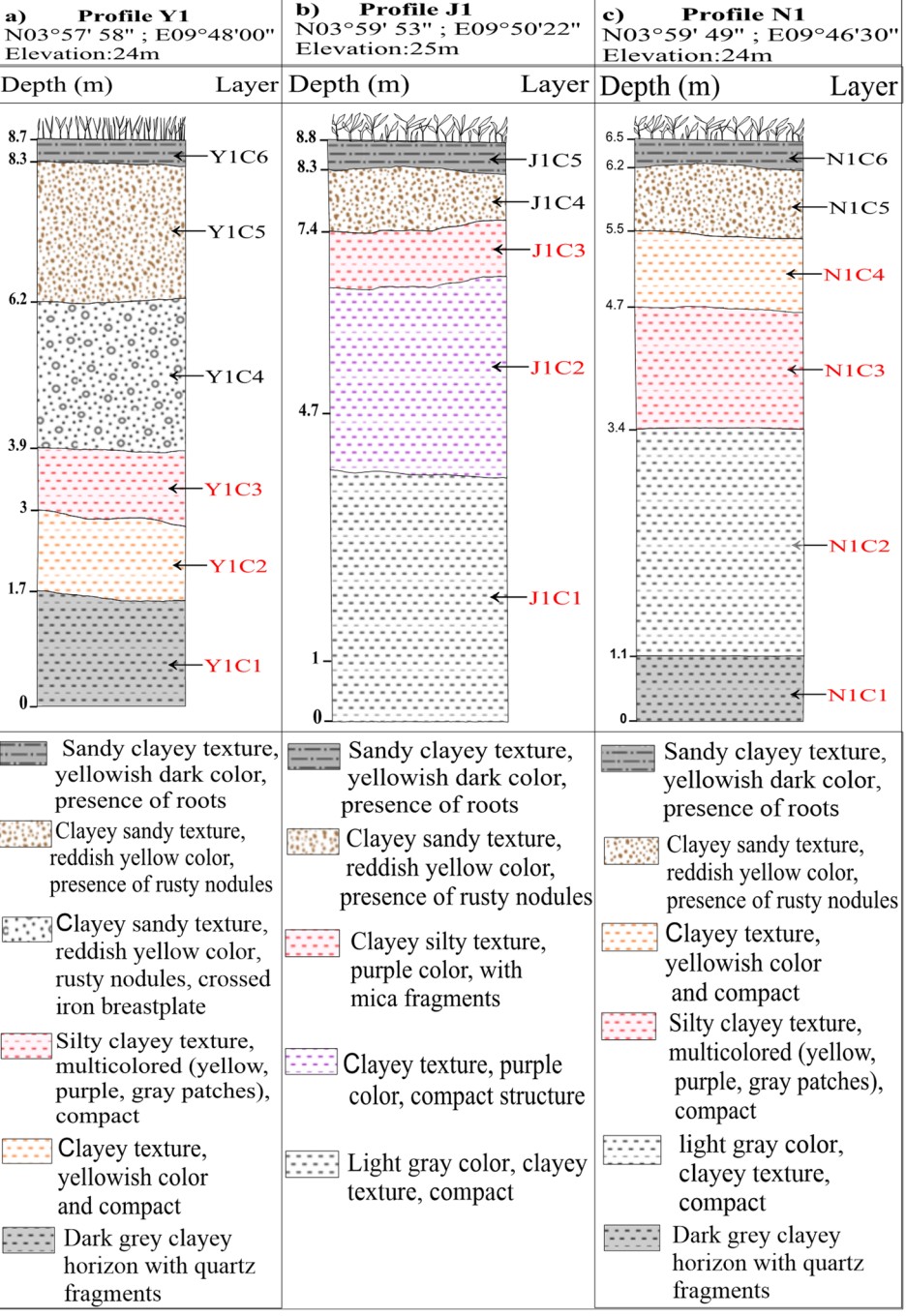

**Figure 2.** Morphology of profiles of the clayey materials of the eastern part of the Douala sub-basin. (**a**) Profile Y1 of Yansoki with analyzed samples Y1C1, Y1C2, Y1C3; (**b**) Profile J1 of Japoma with analyzed samples J1C1, J1C2, J1C3; (**c**) profile N1 of Ndogpassi with analyzed samples N1C1, N1C2, N1C3, N1C4.

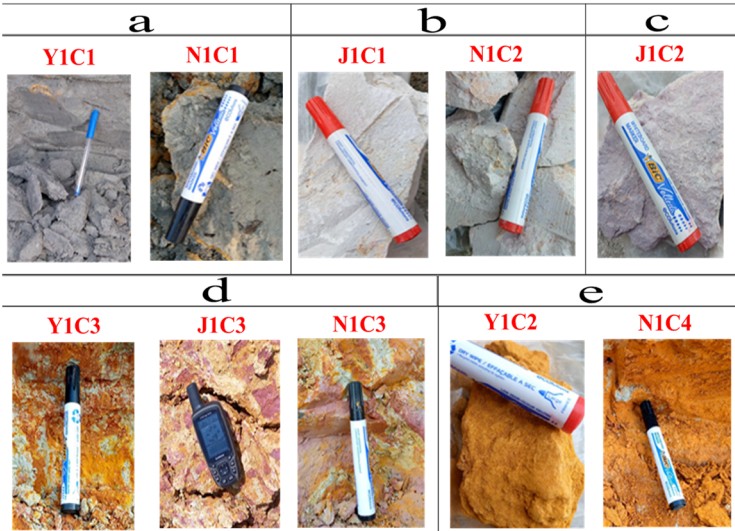

**Figure 3.** Photos of the different facies analyzed in the eastern part of the Douala sub-basin: (**a**) dark gray clay (Y1C1, N1C1); (**b**) light gray clay (J1C1, N1C2); (**c**) purple clay (J1C2); (**d**) multicolored clay (Y1C3, J1C3, N1C3); and (**e**) yellow clay (Y1C2, N1C4).

*3.2. Analytical Techniques*

The mineralogical analyses were carried out at the laboratory AGEs of the University of Liege, Belgium, according to the methodology of Moore Duane and Reynolds Robert [32]. The mineralogy was determined by X-ray powder diffraction on bulk samples and on oriented clay samples at room temperature using a classical powder diffractometer Brüker Advance 8 brand type, equipped with Cu radiation (K$\alpha$ ƙCu $\frac{1}{4}$ 1.54056 Å) at an intensity of 40 mA and a voltage of 40 kV. It was configured with a step size of 0.013° (2$\theta$) for a measurement time of 30 s and data were collected over the interval of 2°–70° (2$\theta$). A mineral phase identification was performed using X'Pert HighScore Plus software associated with PDF-2 2007 release software. Prior to analysis, the analyzed samples were ground and sieved to 80 $\mu$m. The <2 $\mu$m fraction was taken from the suspension after a settling time was calculated according to Stocke's law and it was then placed on a glass slide, meaning the XRD patterns were thus recorded between 2° and 30° 2$\theta$ using the same step size and time per step parameters. These oriented aggregates were subjected to three successive treatments: air drying, glycolation and heating at 500 °C for 4 h, in order to confirm the type of clay mineral phases.

Qualitative and semi-quantitative estimates ($\pm$5–10%) were referred to as the measurements of the maximum intensity of X-ray models according to Biscaye [33]. The infrared spectra were carried out at the Dschang University laboratory in Cameroon. Diffuse reflectance infrared spectra were recorded between 4000 cm$^{-1}$ and 500 cm$^{-1}$ using a Thermo Scientific Nicolet iD7 FTIR spectrophotometer. The spectrum resolution was 4 cm$^{-1}$ and the spectra were obtained by the accumulation of 200 scans.

The geochemical characterization consisted of determining the major element concentrations using the X-ray fluorescence spectrometer (XRF). This analysis was performed on the raw materials using a Niton XL3t980 hXRF analyzer (X-ray tube: 50 kV; anode: silver; silicon detector: 8 mm). The analysis provides raw data in the form of a spectrum with the specific fluorescence energy (in keV) on the x-axis and the photon number (in CPS) on the y-axis. The instrument was calibrated and configured in mining/mineral mode. The materials studied were crushed and sieved to 80 $\mu$m and were then analyzed to obtain the base oxides contained in the materials. The predominance of the oxides was determined by using the triangular diagram $SiO_2$-$Al_2O_3$-$Fe_2O_3$. The evaluation of the level of chemical alterations of the analyzed clays was carried out by the ternary diagram of the evaluation of the chemical alteration index [34]. Weakly altered materials are those with values between

50 and 60%, moderately altered materials are those with values between 60 and 80% and highly altered materials are those with values > 80%. The certainty of the clayey materials in ceramics has been demonstrated using the Fiori and Fabbri diagram [35].

The Chemical Index of Alteration (CIA) was calculated as follows:

$$CIA = (Al_2O_3/(Al_2O_3 + CaO + Na_2O + K_2O)) \times 100 \qquad (1)$$

The physicochemical properties were determined using various methods at Mission de Promotion des Matériaux Locaux (MIPROMALO) in Yaounde, Cameroon. The particle size distribution was determined by a wet sieving method for fractions greater than 80 μm. Particles that were less than 80 μm were analyzed using sedimentation according to the French standard NF P18-056 procedure. The texture of the clay materials was determined using the Belgian textural classification diagram [36]. The porosity and permeability were determined by the McManus triangular diagram [37]. The prediction of the ceramic suitability of kaolinitic clays was examined using a Winkler diagram [38].

The Atterberg limits were used to define the limits of consistency between the solid and plastic state (plasticity limit: PL) and the plastic state to the liquid state (liquid limit: LL). The interval between the plastic limit and liquid limit defines the plastic index (plastic index: PI). The plastic limit test was carried out according to the Casagrande method [39] and the plasticity index (PI) was calculated by the following formula:

$$PI = LL - PL \qquad (2)$$

The binary diagram of Holtz and Kovacs [40] was used to classify the studied clay materials according to their plasticity and the Bain diagram [41] was used as a support to show the suitability of the studied clays in ceramics using the data of the plasticity index of Atterberg: IP < 15% for bricks and IP > 15% for pottery.

The percentage of organic carbon was determined using the Walkley method, which is an oxidation with potassium bicarbonate ($K_2Cr_2O_7$) in an acid medium ($H_2SO_4$). The determination was performed with calorimetry. The amount of organic matter in the clay samples was determined using the amount of organic carbon and the Sprengel coefficient, which is 2. The procedure consists of multiplying the Sprengel factor by the organic carbon content. The swelling property of the clays was determined by adding 20 g of steam powder initially dried at 105 °C for 24 h in a container of volume V1. Subsequently, the powder of the same sample was introduced into a beaker of volume V2 containing 100 mL of distilled water for 24 h at room temperature. The swelling rate (T) is given by the equation:

$$T = 100 \times (V2 - V1)/V1 \qquad (3)$$

The Kjeldahl method [42] was used to determine the cation exchange capacity (CEC) with a three-step protocol: saturation of the absorbent complex by the addition of $NH_4^+$ ions and the extraction of exchangeable bases; washing of the clay with ethanol to remove excess $NH_4^+$ ions; and determination of $NH_4^+$ by distillation. The procedure consists of adding two drops of phenolphthalein in a 10 mL extract, then adding soda until a pink–purple coloration appears. Distilled, the $NH_3$ is trapped in boric acid and is titrated with $H_2SO_4$ 0.01N using a potassium chloride solution (KCl) as a developer.

## 4. Results and Discussion

### 4.1. Mineralogy

The results obtained from the analysis of the X-ray diffraction of the bulk clay materials showed the same mineralogical phases with varying concentrations (Figure 4). The mineralogical composition of the studied samples included quartz ($SiO_2$), kaolinite ($Al_2(OH)_4Si_2O_5$), illite ($(K,H_3O(Al,Mg,Fe)_2(Si,Al)_4O_{10}[(OH)_2,(H_2O)])$), hematite ($Fe_2O_3$), muscovite ($KAl_2(AlSi_3O_{10})(F,OH)_2$), feldspar ($KAlSi_3O_8$–$NaAlSi_3O_8$–$CaAl_2Si_2O_8$), goethite (FeOOH) and gibbsite ($Al(OH)_3$). Figure 5 presents the XRD patterns of the <2 μm frac-

tion of the analyzed samples and Table 1 estimates the semi-quantitative mineralogical composition of the mineral phases of the clay materials based on the peak intensities of the XRD patterns. This revealed an important proportion of kaolinite (48.3–69.2 wt.%), quartz (20.5–41.2 wt.%) and appreciable amounts of illite (2.4–10.2 wt.%), which are in relation. In Ndogpassi, kaolinite was the most abundant mineral followed by quartz. In Yansoki and Japoma, kaolinite was also the main mineral, but in lower proportions here than in Ndogpassi. Illite remained the same in Ndogpassi. This clay mineral was shown to be present by the less intense peaks at 10.3 Å and 3.34 Å, which indicate low crystallinity. Illite is known for its cohesive, swelling and plastic properties as ceramic products [43,44]. The Y1C2 and N1C4 samples stood out from other samples due to the presence of hematite (3.5 and 3.8%, respectively), which at these proportions gave a yellow color to these samples [19,20]. The amounts of the other mineral phases were: hematite (0–3.8 wt.%), goethite (0–0.8 wt.%), feldspar (0–3.4 wt.%), muscovite (0–2.7 wt.%) and gibbsite (0–1.8 wt.%), respectively. These identified minerals are characteristic of marine deposits in Cameroon's coastal basins [28,45]. In Figure 5, kaolinite is characterized by diffraction peaks between 7.11 and 7.18 Å in the normal state (N) and after treatment with ethylene glycol (EG), which then disappears after heating to 500 °C (CH). This clay mineral had a sharp diffraction peak, indicating that it is crystalline. However, illite was indicated by low peaks between 9.56 and 10.3 Å at normal (N) and did not change after being treated with ethylene glycol (EG) and heated to 500 °C (CH). These minerals are mainly accompanied by quartz. According to Hinckley [46], kaolinite and quartz peaks on the diffractograms are well expressed, indicating that the minerals are well crystallized. However, other similar peaks were identified in the raw clay materials with broad peaks or were weakly expressed, which is characteristic of minerals with a lower degree of crystallinity. The mineralogical composition of the samples in this study is similar to that of the Douala clay materials [25,26]. However, the results obtained were compared with other clay materials that were formed under the same conditions, notably the raw clay materials of Ketou Benin [47], Lokoundje Cameroon [48] and Tamazert, Hadj Ali and Chekfa from Algeria [49]. The same mineralogical characteristics of the mineral assemblage were observed, except the rutile concentration was found in Lokoundje Kribi Cameroon [48]. The results from the XRD analysis can be confirmed by FTIR spectra (Figure 6), indicating the presence of kaolinite, quartz, illite and oxyhydroxide minerals. The absorption peak found between 3700 $cm^{-1}$ and 3600 $cm^{-1}$ (3692.60 $cm^{-1}$, 3615.23 $cm^{-1}$) in all samples attests the presence of kaolinite [28,48]. The lack of a well-defined peak at 3664 $cm^{-1}$ may be due to the existence of disordered kaolinite [50]. The peaks of quartz found between 1200 $cm^{-1}$ and 900 $cm^{-1}$ (1115.98 $cm^{-1}$, 1006.31 $cm^{-1}$, 904 $cm^{-1}$) for all the samples because of the stretching vibration of Si-O confirms the presence of this mineral and corroborates with the results from XRD analysis [51]. However, the absorption at 3364.33 $cm^{-1}$ and 740 $cm^{-1}$ indicates the presence of illite [52]. These characteristics confirmed the origin of the sediments of the studied samples [50]. According to [52], water molecules are revealed by the wide band at 1632 $cm^{-1}$ due to the H–O–H vibrations of the adsorbent. In this work, the band at 1632.81 $cm^{-1}$ or 1612 $cm^{-1}$ in all samples is attributed to the absorption of the water molecule (OH), i.e., the presence of hydrated phyllosilicates. This progressive decreasing of the band is related to the removal of octahedral cations, causing the loss of water. The presence of kaolinite, quartz and illite in the material gives it a better capacity in the production of ceramics and terracotta due to their refractory properties at a liquidus temperature of about 1800 °C [4,53]. Accordingly, the mineralogical compositions are suitable for ceramics products.

**Table 1.** Semi-quantitative estimation of the mineralogical composition (%) of clay materials of the eastern part of the Douala sub-basin based on peak intensities of XRD patterns.

| Localities | Yansoki | | | Japoma | | | Ndogpassi | | | |
|---|---|---|---|---|---|---|---|---|---|---|
| Samples | Y1C1 | Y1C2 | Y1C3 | J1C1 | J1C2 | J1C3 | N1C1 | N1C2 | N1C3 | N1C4 |
| Kaolinite | 50.5 | 52.3 | 49.5 | 50.4 | 48.3 | 51.4 | 54.7 | 69.2 | 50.1 | 49.8 |
| Quartz | 40.8 | 39.2 | 41.2 | 35.6 | 39.4 | 38.6 | 26.4 | 20.5 | 31.4 | 33.4 |
| Illite | 3.9 | 3.1 | 4.7 | 2.4 | 8.1 | 3.4 | 8.4 | 3.4 | 9.7 | 10.2 |
| Hematite | 1.4 | 3.5 | - | 2.9 | - | 2.7 | 2.5 | - | 3.8 | 2.6 |
| Goethite | - | 1.9 | 2.4 | 3.5 | 3.2 | 1.4 | 1.7 | 3.6 | 3.3 | 2.3 |
| Feldspar | 2.5 | 1.8 | - | 1.6 | - | - | 3.4 | 1.2 | - | 0.5 |
| Muscovite | 1.9 | 1.4 | 2.2 | 1.8 | - | 1.2 | 2.7 | 1 | - | - |
| Gibbsite | - | 0.6 | - | 1.8 | 1 | 1.3 | 1.8 | 1.1 | 1.7 | 1.2 |

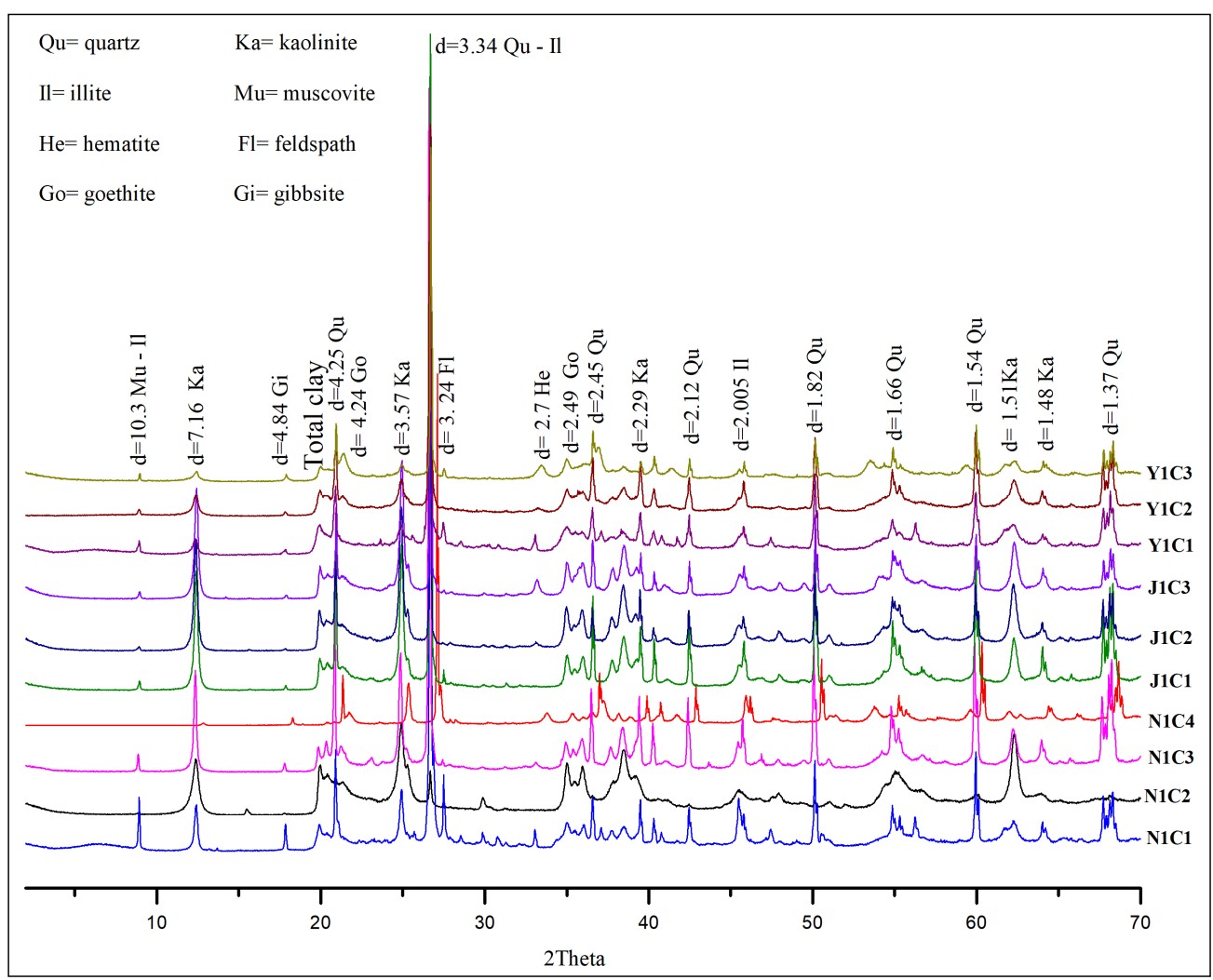

**Figure 4.** XRD patterns of raw clay materials of the eastern part of the Douala sub-basin.

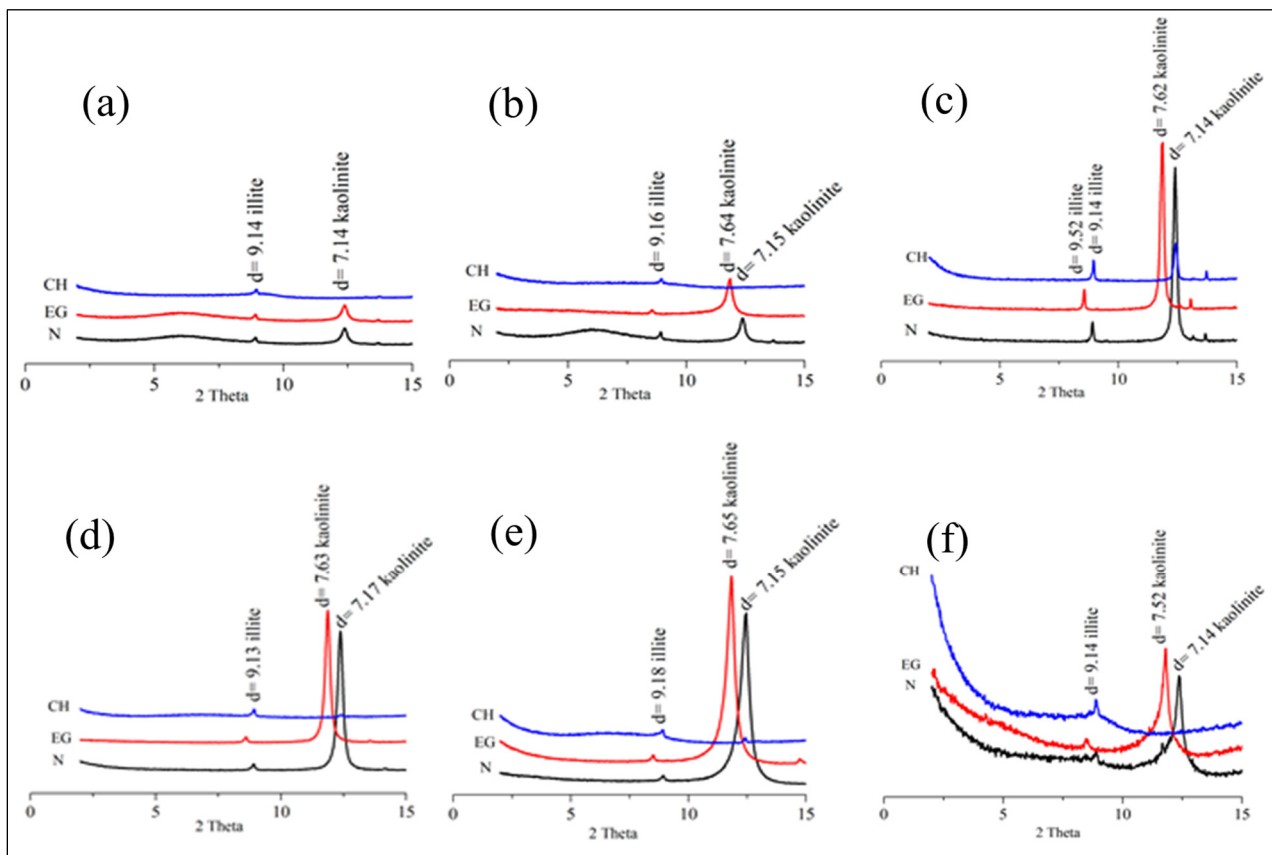

**Figure 5.** XRD patterns of the clay fraction: (**a**) Y1C1; (**b**) Y1C3; (**c**) J1C2; (**d**) J1C3; (**e**) N1C2 and (**f**) N1C4. N: normal sample, CH: heated at 500 °C, EG: treated with ethylene glycol.

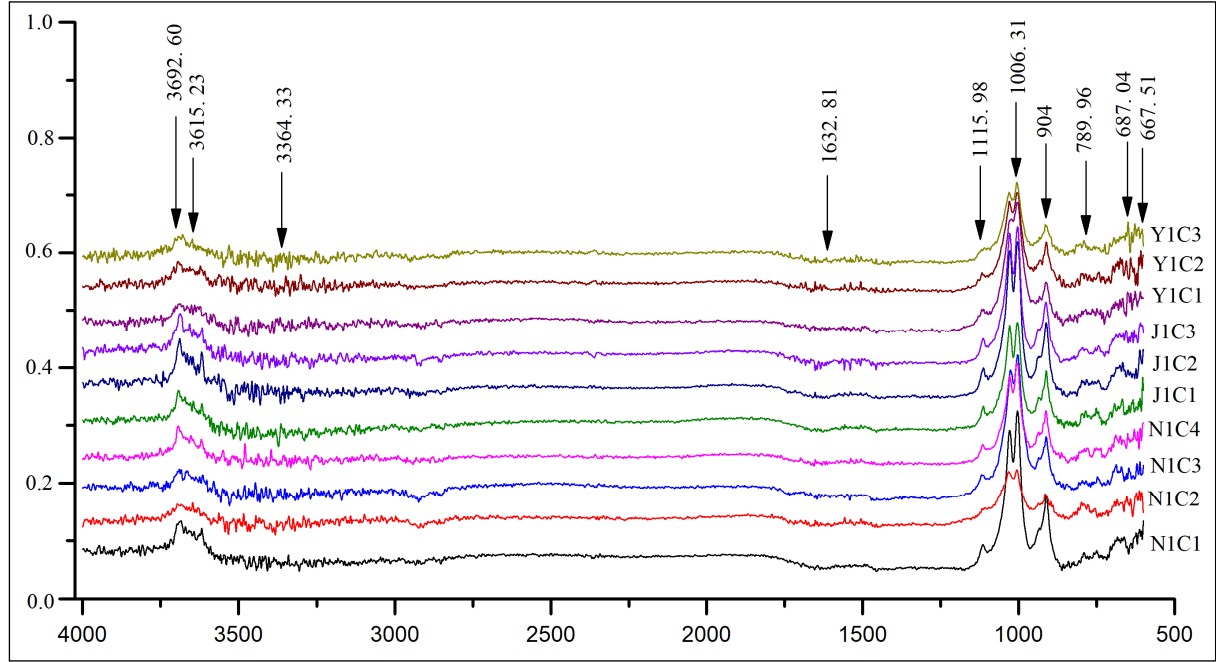

**Figure 6.** IR curves of the samples of the eastern part of the Douala sub-basin.

*4.2. Geochemistry*

The results of the XRF chemical analyses expressed as mass percentages of the oxides of the selected samples from different sites of the study area are presented in Table 2. The geochemical analysis of clay materials gives a precise idea of their origin and the processes that contributed to their formation, but also provides the information necessary to understand their technological properties [54]. The observation of the geochemical composition presented shows that the studied samples mainly consisted of silica ($SiO_2$), alumina ($Al_2O_3$), iron oxide ($Fe_2O_3$) and traces of $TiO_2$, CaO, MgO, $Na_2O$, $K_2O$ and $SO_3$. The samples from Yansoki basically contained 38.23–51.18% of silica, 13.01–16.45% of alumina and 4.9–17.76% of iron oxide. The samples from Japoma contained both the highest (58.03%) and lowest (38.18%) fraction of silica. The alumina content was 19.92–22.94% and 3.86–6.43% of iron oxide. The contents of the oxides in the samples from Ndogpassi were constituted with 22.21–55.62% of silica, 12.84–19.37% of alumina and 1.07–17.92% of iron oxide. These oxides were in preponderant proportions in several studied samples of Cameroonian coastal basins [25,48,55]. The occurrence of CaO, MgO, $Na_2O$ and $K_2O$ was low, which indicates the unlikely presence of high swelling clay minerals such as montmorillonite, except for the N1C4 sample which had 11.7% of $K_2O$ [55,56]. The high percentage of $SiO_2$ (22.21–58.03%) demonstrates the abundance of quartz in the samples. $Al_2O_3$ (12.84–22.94%) may be linked to the presence of kaolinite. The iron oxide content (1.07–17.92%) explains the existence of hematite and goethite in the clays and consequently are at the origin of the reddish brown or reddish yellow coloring (Y1C2 and N1C4) [45,55]. In view of these analyses, the geochemical composition is in agreement with the mineralogical data and effectively confirms the presence of the different mineral phases that have crystallized, such as quartz and kaolinite. The ratio of $SiO_2/Al_2O_3$ varied between one and three, except for samples such as Y1C1 (3.11), N1C3 (4.33) and N1C1 (3.22) (Table 3). According to [57,58], the low $SiO_2/Al_2O_3$ ratios close to two in the material demonstrates the predominance of type 1:1 clays. Thus, the $SiO_2/Al_2O_3$ ratios of most of the samples studied were close to two, which would indicate the presence of minerals of the kaolinite family (type 1:1). The N1C3, N1C1 and Y1C1 samples had ratios that were greater than two, suggesting the presence of free forms of silica and clay minerals type 2/1, such as illite in the materials as well as a chemical maturity of the studied clay materials [57,58]. Low levels of the exchangeable cations CaO, $Na_2O$ and $K_2O$ observed in all the samples confirm the low content of the 2:1 type mineral [56]. Therefore, the clay materials in the study area had significant proportions of type 1:1 clay minerals compared to type 2:1 clay minerals. A representation of the $SiO_2$-$Al_2O_3$-$Fe_2O_3$ triangular diagram (Figure 7) showed that all samples were located towards the $SiO_2$ pole and along the $SiO_2$-$Al_2O_3$ axis, which is in agreement with a $SiO_2/Al_2O_3$ ratio. This indicates excesses of $SiO_2$ and $Al_2O_3$ and effectively confirms the presence of quartz and kaolinite in the studied samples [57,58]. For most selected samples from different studied sites, the variation of LOI (between 8.19 and 16.11%) reveals the presence of mainly hydrated minerals such as clay and goethite [59]. These results would indicate the presence of minerals as halloysite with a high hydration capacity. The Chemical Index of Alteration (CIA) varied from 61.59–97.76%, suggesting that the studied clays underwent an intense alteration (Table 2). This result is in perfect harmony with the morphological and climatic conditions prevailing in the coastal region [24,60]. The geochemical analyses were projected in the ternary diagram of Nesbitt and Young [34] (Figure 8). The positions of the samples in the diagram showed that the materials had a high degree of chemical alteration except for the N1C4 sample, which was in the moderate weathering range. The higher values of the chemical weathering indices in the materials reflect further weathering due to the climatic conditions of the study sites [24,26,60]. The suitability of the raw clay materials for the use in ceramic industries is demonstrated by the application of ternary diagram [35] (Figure 9). The projection of the samples in this diagram presents a suitable composition for ceramic production and terracotta building materials. These clay materials can be used in the production of sandstone tiles and stoneware.

**Table 2.** Geochemical composition (wt.%) of clay materials of the eastern part of the Douala sub-basin determined by XRF.

| Localities | Yansoki | | | Japoma | | | Ndogpassi | | | |
|---|---|---|---|---|---|---|---|---|---|---|
| Samples | Y1C1 | Y1C2 | Y1C3 | J1C1 | J1C2 | J1C3 | N1C1 | N1C2 | N1C3 | N1C4 |
| $SiO_2$ | 51.18 | 38.97 | 38.23 | 58.03 | 38.18 | 49.97 | 50.07 | 22.21 | 55.62 | 48.23 |
| $TiO_2$ | 1.09 | 0.06 | 0.75 | 1.68 | 1.47 | 1.43 | 1.13 | 0.98 | 1.16 | 2 |
| $Al_2O_3$ | 16.45 | 15.27 | 13.01 | 19.92 | 21.85 | 22.94 | 15.54 | 17.32 | 12.84 | 19.37 |
| $Fe_2O_3$ | 4.9 | 17.76 | 5.42 | 3.88 | 3.86 | 6.43 | 4.22 | 1.07 | 17.92 | 6.06 |
| MnO | 0.41 | 0.12 | 0.13 | 0.33 | 0.07 | 0.09 | 0.14 | 0.82 | 1.02 | 0.62 |
| MgO | 0.95 | 0.32 | 0.43 | 0.3 | 0.1 | 0.2 | 0.74 | 0.15 | 0.32 | 0.53 |
| CaO | 0.72 | 0.08 | 0.23 | 0.04 | 0.06 | 0.21 | 0.15 | 0.15 | 0.38 | 0.27 |
| $K_2O$ | 1.7 | 0.51 | 0.48 | 0.85 | 0.41 | 0.42 | 2.24 | 0.34 | 1.62 | 11.7 |
| $Na_2O$ | 0.06 | 0.02 | 0.34 | 0.37 | 0.03 | 0.2 | 0.08 | 0.01 | 0.19 | 0.11 |
| $SO_3$ | 0.08 | 0.68 | 0.13 | 0.09 | 0.15 | 0.27 | 0.32 | 0.34 | 0.51 | 0.02 |
| $P_2O_5$ | 0.13 | 0.11 | 0.05 | 0.05 | 0.06 | 0.07 | 0.02 | 0.16 | 0.02 | 0.17 |
| LOI | 13.2 | 12.77 | 11.21 | 8.37 | 11.23 | 10.06 | 16.11 | 15.4 | 8.19 | 8.58 |
| Total | 90.87 | 86.67 | 70.41 | 93.91 | 77.47 | 92.29 | 90.76 | 58.95 | 99.79 | 97.75 |
| $SiO_2/Al_2O_3$ | 3.11 | 2.55 | 2.94 | 2.91 | 1.75 | 2.18 | 3.22 | 1.28 | 4.33 | 2.49 |
| $TiO_2 + Fe_2O_3$ $+ CaO + MgO$ $+ Na_2O + K_2O$ | 9.42 | 18.75 | 7.65 | 7.12 | 5.93 | 8.89 | 8.56 | 2.7 | 21.59 | 20.76 |
| CIA | 86.89 | 96.16 | 92.53 | 94.05 | 97.76 | 96.51 | 86.28 | 97.19 | 85.43 | 61.59 |

**Table 3.** Physicochemical parameters of the studied clays of the eastern part of the Douala sub-basin.

| | | YANSOKI | | | JAPOMA | | | NDOGPASSI | | | |
|---|---|---|---|---|---|---|---|---|---|---|---|
| Samples Codes | | Y1C1 | Y1C2 | Y1C3 | J1C1 | J1C2 | J1C3 | N1C1 | N1C2 | N1C3 | N1C4 |
| Texture | | Silty clay | Silty clay | Silty clay | Very heavy clay | Very heavy clay | Very heavy clay | Heavy clay | Very heavy clay | Silty clay | Very heavy clay |
| Color | | Dark gray | Yellow | Multicolored | Light gray | Purple | Multicolored | Dark gray | Light gray | Yellow | Multi-colored |
| Thickness (m) | | 1.7 | 0.7 | 0.9 | 3.7 | 2.7 | 2.7 | 2 | 2.3 | 0.8 | 0.8 |
| Sand (200 > Φ > 20 μm) % | | 0.9 | 1 | 0 | 1 | 0.8 | 1.5 | 0.9 | 0 | 4 | 0,2 |
| Silt (20 > Φ > 2 μm) % | | 68.1 | 68.5 | 71.5 | 41.5 | 40 | 30.5 | 65.1 | 1 | 70 | 43.3 |
| Clay (Φ < 2 μm) % | | 31 | 30.5 | 39.5 | 65.5 | 59.2 | 68 | 34 | 99 | 26 | 56.5 |
| Liquid limit (LL) | | 73.3 | 72.2 | 70.6 | 36.5 | 50.5 | 38.8 | 73.7 | 109.2 | 46.7 | 62.6 |
| Plastic limit (PL) | | 50.4 | 53 | 49.5 | 22.5 | 27.1 | 27.7 | 47.3 | 45.3 | 36.5 | 43.8 |
| Plasticity index (PI) | | 21.9 | 19.2 | 21.1 | 14 | 23.4 | 11.1 | 26.4 | 63.9 | 10.2 | 18.8 |
| Organic matter (%) | | 4 | 1.19 | 1.07 | 2.3 | 2.04 | 1.02 | 6.29 | 2.06 | 0.19 | 2.53 |
| Sweling rate (T) | | 13.6 | 11.4 | 9.1 | 5 | 10 | 5 | 11.4 | 13 | 7.5 | 11.4 |
| CEC (meq/100 g) | | 27.12 | 23.18 | 17.09 | 21.30 | 19.74 | 16.59 | 27.45 | 26 | 26.5 | 38.12 |
| pH | pH-$H_2O$ | 3.4 | 4.8 | 4.7 | 5 | 5 | 4.9 | 2.8 | 4.7 | 5.3 | 4.5 |
| | pH-KCl | 2.9 | 4.2 | 3.8 | 4.1 | 4.2 | 3.9 | 2.6 | 3.9 | 4.7 | 4 |

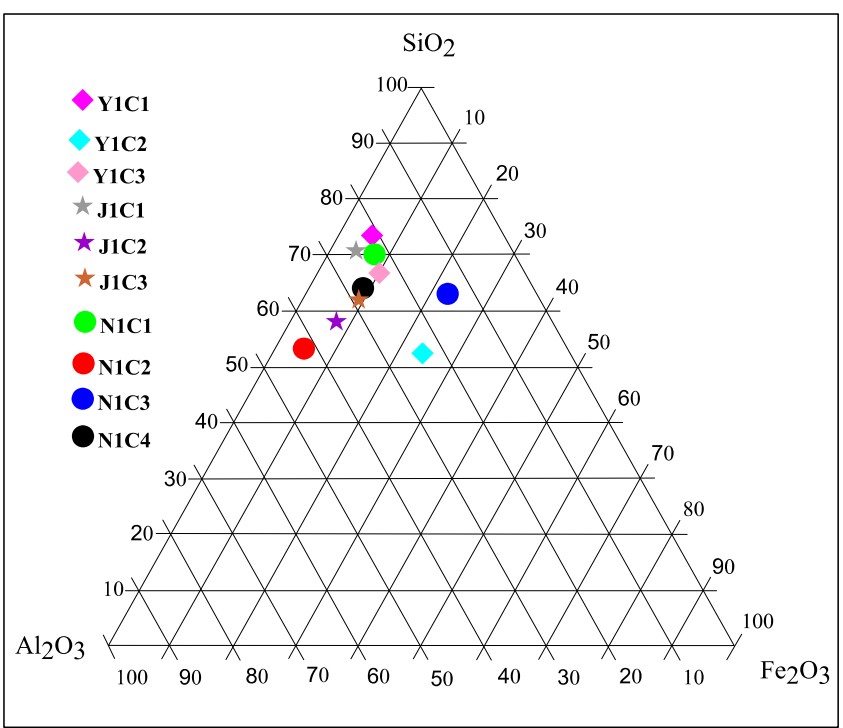

**Figure 7.** Position of clay materials of the eastern part of the Douala sub-basin in the triangular diagram SiO$_2$-Al$_2$O$_3$-Fe$_2$O$_3$.

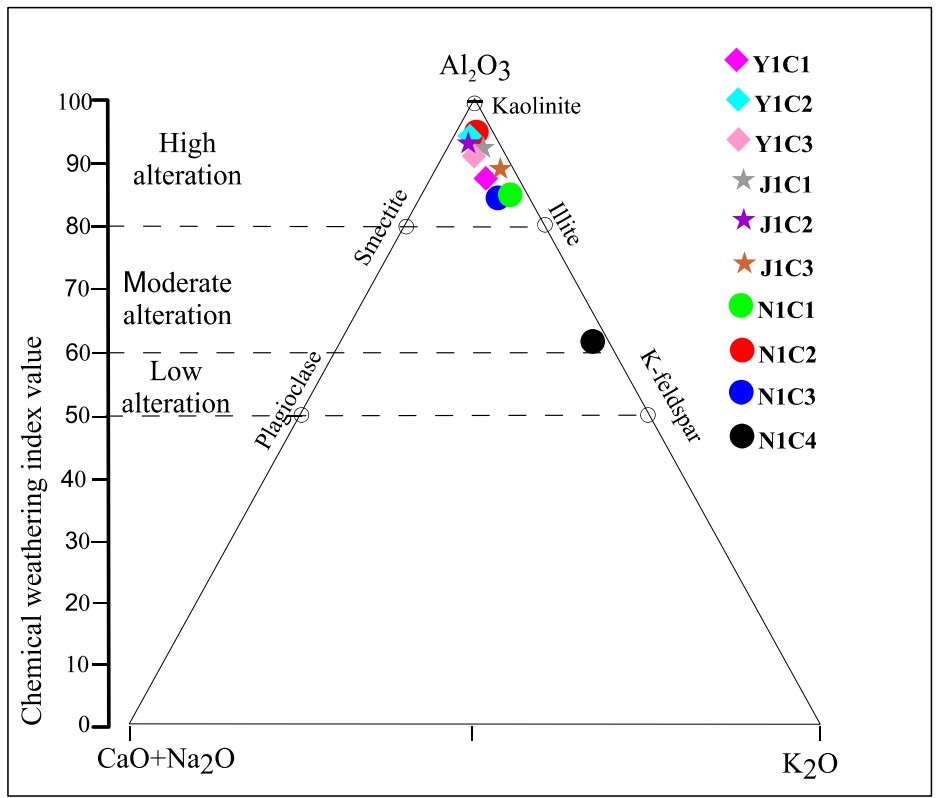

**Figure 8.** Position of the studied clayey materials in the ternary diagram of Nesbitt and Young [34].

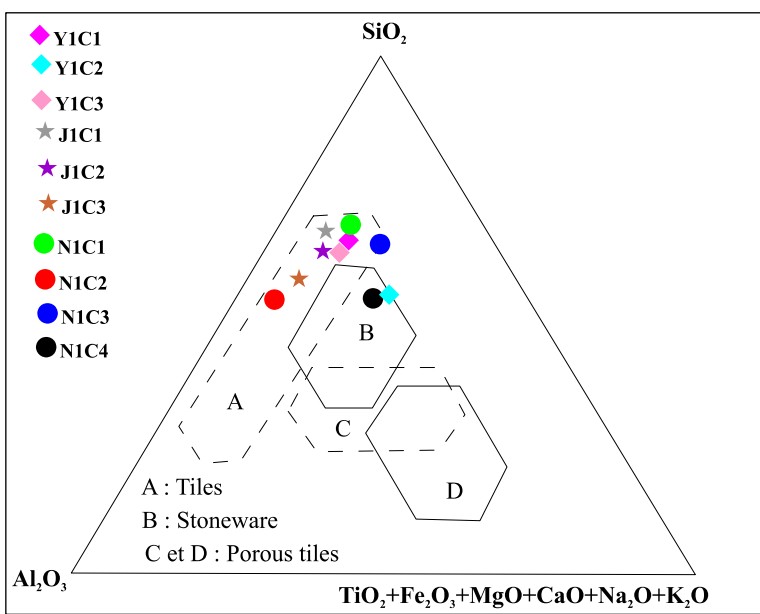

**Figure 9.** Composition of clay materials of the eastern part of the Douala sub-basin in the diagram of Fiori et al. [35].

### 4.3. Physicochemical Properties

#### 4.3.1. Particle Size Fractions

Knowledge of the particle size distribution of clay materials is an important factor to determine its suitability for various applications [1]. Table 3 presents the physicochemical characteristics of the different samples. Generally, the different granulometric proportions of the analyzed samples showed that the clay fraction (<2 μm) ranged from 26% (N1C3) to 99% (N1C2), silt (2 < Ø > 20 μm) and sand (20 < Ø > 200 μm) varied from 1% (N1C2) to 70% (N1C3) and 0 (Y1C3, N1C2) to 4% (N1C3), respectively. The proportions of the clay fraction (<2 μm) varied from 30.5% to 39.5% for Yansoki, 59.2–68% for Japoma and 34–99% for Ndogpassi. The clay fraction had the most quantities in all the studied samples (50.9%, average amount). The proportions of the silt fraction (2 < Ø > 20 μm) ranged between 68.1% to 71.5% for Yansoki, 30.5% to 41.5% for Japoma and 1% to 70% for Ndogpassi. Except for the N1C1 sample, the studied samples from Yansoki and Ndogpassi had higher quantities of the silt fraction. Silt was the second most important fraction of the studied samples. However, the quantities of the sand fraction were 0% to 1% in Yansoki, 0.8–1.5% in Japoma and 0–4% in Ndogpassi. The sand fraction had the lowest quantities in the selected samples from different sites of the study area. These samples were therefore richer in clay fractions (<2 μm) than the clay materials of the Lake Chad basin, which ranged from 16.54 to 21.54% [1]. The higher content of fine fraction (clay and silt) observed in the clay materials is favored by morphoclimatic and hydrological conditions in the Littoral region of Cameroon. The hot and humid tropical climate of the Littoral region favors the intense alteration of the materials compared to the clay materials of the Lake Chad basin, which are subjected to the sahelo-sudanian and sahelo-saharan climates. The particle size distribution of clay plays an essential role in defining the properties of suspensions (plasticity and viscosity) and green pastes during drying and firing [19,61].

The particle size analysis of the studied samples was plotted in the Belgian diagram of textural classifications of clayey materials [36]. This diagram indicates that the clay materials are very heavy clays and silty clays (Figure 10). This is in perfect correlation with the most amount of fine particle sizes found in the areas corresponding to very heavy clays and silty clays.

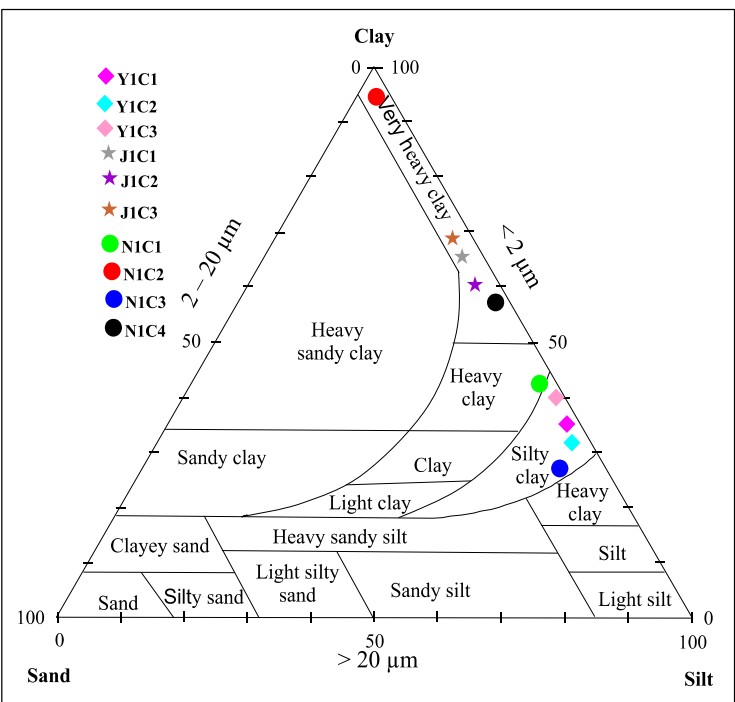

**Figure 10.** Classification of the clay materials of the eastern part of the Douala sub-basin according to the Belgian triangle of textures [36].

The ternary diagram of McManus [37] was used to classify clay materials according to their porosity and permeability. According to this diagram [37], all the studied samples may be classified as weakly porous and permeable with the exception of the N1C2 sample from Ndogpassi, which was found in the range of high porosity and very low permeability (Figure 11). The permeability therefore depends largely on the shape, size and sorting of the particles. This observation agrees with the distribution of the constitutive particles of the studied clayey materials.

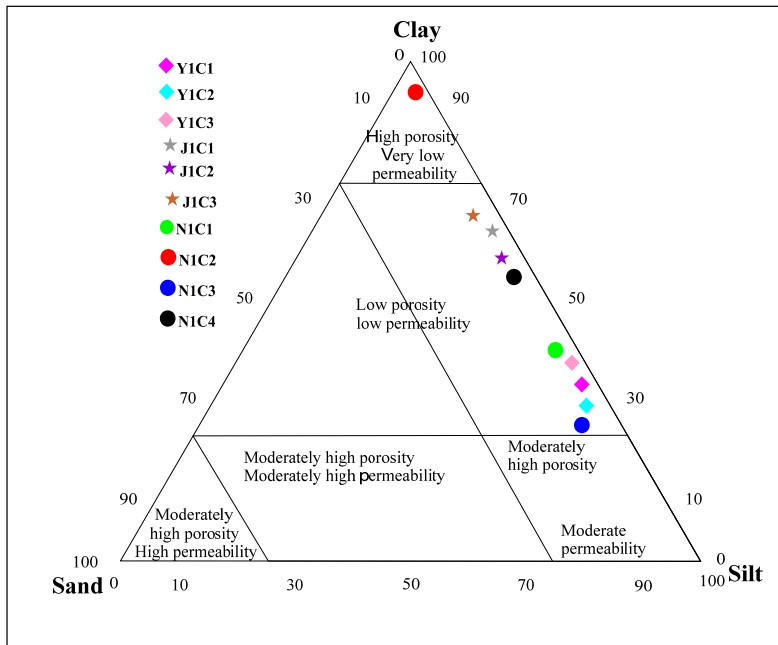

**Figure 11.** Porosity and permeability of clay materials of the eastern part of Douala sub-basin in the triangular diagram after [37].

The particle size was projected in the Winkler diagram [38] (Figure 12). The use of the Winkler diagram made it possible to predict the ceramic suitability of clay materials. However, neither the firing conditions nor the properties of ceramic magnitudes can be known. The particle size distribution of the eastern part of the Douala sub-basin showed that they are made up of fine particles. In the Winkler diagram, the results showed that the studied clayey materials are good candidates and can be used for the production of different bricks and tiles if they are mixed with 10% of coarse size particles such as sand in the manufacturing process [38].

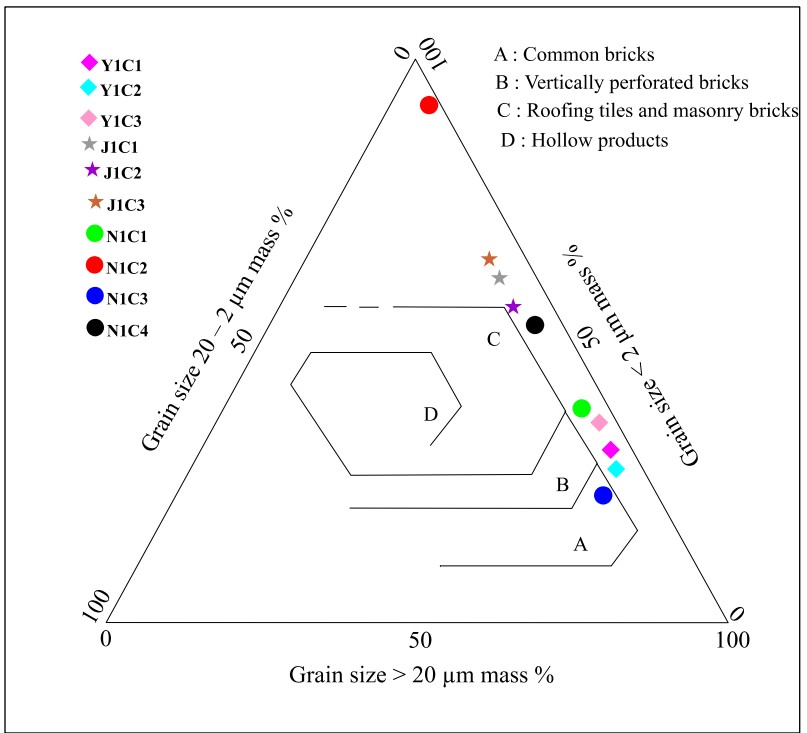

**Figure 12.** Suitability of the studied clayey materials according to the Winkler ternary diagram [38].

### 4.3.2. Plastic Properties

Plasticity is a very important parameter for manufacturing clayey products [15]. The Atterberg limits are presented on Table 3 and are plotted on the Holtz and Kovacs diagram (Figure 13). The limits of plasticity ranged between 49.5 (Y1C3) and 53 (Y1C2) for Yansoki, 22.5 (J1C1) and 27.7 (J1C3) for Japoma and 36.5 (N1C3) to 47.3 (N1C1) for Ndogpassi. The plasticity index varied from 19.3 (Y1C2) to 21.9 (Y1C1), 11.1 (J1C3) to 23.4 (J1C2) and 10.2 (N1C3) to 63.9 (N1C2), respectively, for Yansoki, Japoma and Ndogpassi sites. Globally, the plasticity of the studied samples was mostly influenced by the clay fraction. Hence, the N1C2 sample had a high clay amount and high plasticity index value (63.9), but the N1C3 sample had the lowest plastic index (10.2) with less clay proportion. The Holtz and Kovacs diagram [40] was used to classify the clay soils on a scale of four plasticity domains (Figure 13): non-plastic clay soils, low plasticity, medium plasticity and high plasticity. It was also used to account for the mineralogical nature that was predominant in the clay materials. On the diagram, the samples fall within the zone of high plastic clays (Y1C1, Y1C2, Y1C3, N1C2, N1C1 and N1C4) and medium plastic clay (J1C1, J1C2, J1C3 and N1C3). The liquidity limits of these materials are high and while those of an average plasticity are around 40.31%. This suggests the hygroscopic nature of the materials. Plasticity is favorable for extrusion and manual processing due to the proportion of clays and silts. It is consistent with the classification from the Winkler diagram and confirms the clay and silt content of the studied samples (Figure 12). Based on the Holtz and Kovacs diagram [40], the observations are in agreement with the results of the granulometric tests. According

to Bukalo [55], the plasticity of the samples from the eastern part of the Douala sub-basin corresponds to that of the coastal basins of Cameroon.

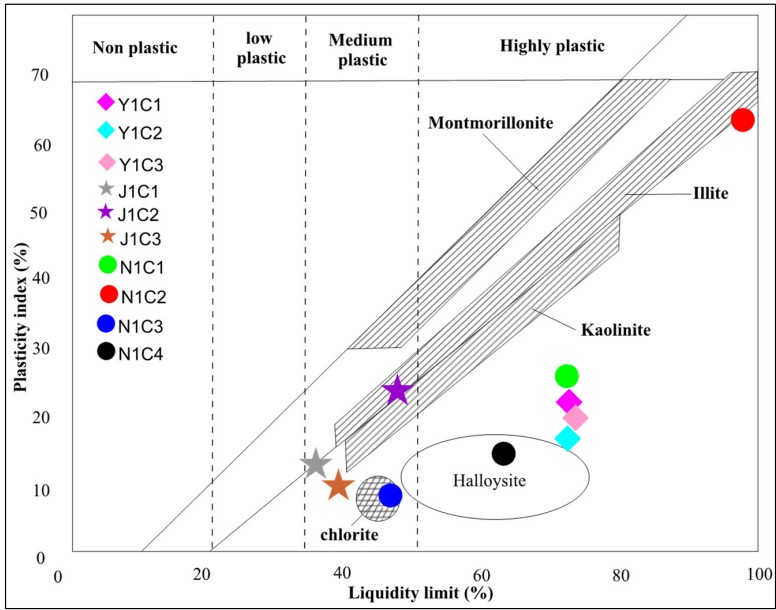

**Figure 13.** Binary diagram of Holtz and Kovacs [40]: correlation between plasticity and mineralogy of clayey materials of the studied area.

Numerous studies have shown that particle size distribution and clay content have an influence on the plasticity of clay raw materials and therefore influence rheological properties [40,62]. The Atterberg limits of the samples are reported in the Bain diagram [41] presented in Figure 14. The use of this diagram makes it possible to predict the suitability of clay materials for ceramic production. Therefore, the Bain diagram [41] shows that these clay samples can be used to manufacture bricks (N1C3, J1C3 and J1C1) or pottery for the rest of the studied samples (Y1C1, Y1C2, Y1C3, J1C2, N1C2, N1C1 and N2C3). Generally, the samples of the study area may be more improved if they are mixed with coarse particles such as sand. The plasticity index (PI) values of the studied samples are higher than 10%, which implies that these samples are appropriate for building ceramic products due to the appropriate extrusion process [41].

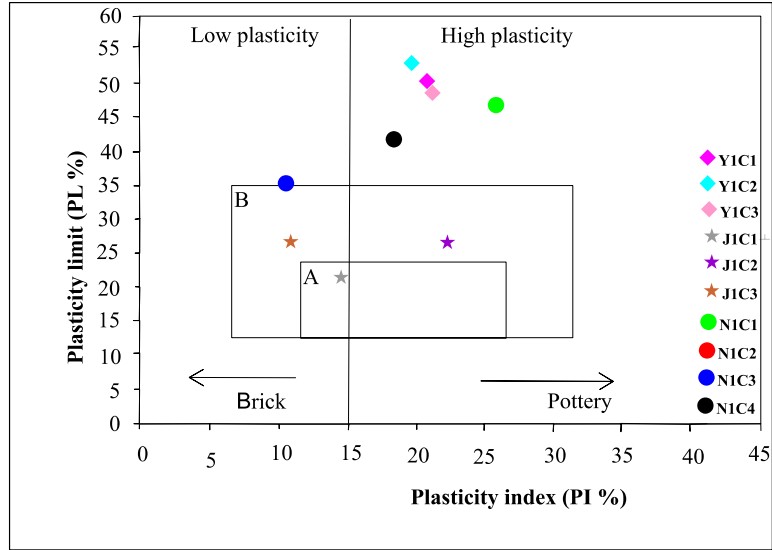

**Figure 14.** Potential molding of the analyzed clayey materials according to Bain diagram [41].

### 4.3.3. Organic Matters

The amount of organic matter (OM) in the studied samples ranged from 0.19% (N1C3) to 6.29% (N1C1) and these results are presented in Table 3. The average of the amount of organic matter was 2.27% and it was moderate. However, the organic matter has no effect on the final ceramic product when particles sizes are fine, but when the particle sizes are large, they show wide pores and carbon marks on the product after firing [63]. According to Wilding [64], the dark coloration of some clayey layers observed on the described and exploited profiles can come from the appreciable content of organic matter which forms with the clay, the clay–humic complex. Thus, the samples Y1C1 and N1C1 are the witnesses of the presence of organic matter in the clayey materials.

### 4.3.4. Rate of Swelling

The swelling index is an essential property of clays to understand their swelling or non-swelling character and to give indications on the nature of the clay materials. According to Bukalo [40], fine particle materials need a large amount of water to become a paste and subsequently become liquid. Thus, the material tends to retain more water, which leads to the elevation of the swelling rate and plasticity index. The results showed that the swelling rate in the analyzed samples ranged from 5 to 13.6 (Table 1). This rate therefore remains low to medium. This is in line with the non-hygroscopic character of the type 1/1 clayey minerals, indicated by the XRD showing the kaolinitic nature.

### 4.3.5. Cation Exchange Capacity (CEC) and the Hydrogen Potential (pH)

The cation exchange capacity (CEC) values of the studied clayey materials ranged between 16.59 and 38.12 meq/100 g (Table 3). These values resulted from the kaolinite phase and are coherent with the mineralogical and chemical composition of the studied samples [42]. The presence of clay minerals such as illite and mica accounted for these cation exchange capacity values [65]. In addition, the cationic exchange capacity did not exceed 28 meq/100 g. This justifies the presence of non-swelling minerals such as kaolinite, illite and muscovite identified by XRD.

The clay samples had acidic pH values (Table 3). These values ranged between 2.8 and 5.3 for pH-$H_2O$ and 2.6 and 4.7 for pH-KCl. This result confirms the absent of clay minerals such as calcite, as already revealed by XRD.

## 5. Conclusions

In the eastern part of the Douala sub-basin, the exploratory field trip (Yansoki, Japoma and Ndogpassi) exhibited good indices of clay materials for ceramics, with a thickness of more than 7 m over an area of the sub-basin. The geochemical, mineralogical and physicochemical characterizations were the investigated methods on clay materials. The morphological description of the raw clay deposit is made up of various clay facies (dark gray, light gray, purple, yellow and multicolored). Mineralogically, the main clay minerals of the studied samples consisted of kaolinite (48.3–69.2 wt.%) and quartz (20.5–41.2 wt.%), with lesser amounts of illite content (2.4 to 10.2 wt.%). Other minerals were also observed, such as hematite, goethite, feldspar, gibbsite and micas, but in small to moderate amounts. This mineralogical result suggested that monosiallitization is the main crystallochemical process acting in the study area. Geochemical results showed that $SiO_2$ (22.21–58.03%) and $Al_2O_3$ (12.84–22.94%) are the main oxides with lesser to higher amounts of $Fe_2O_3$ content (1.07–17.92%). The significant amount of these oxides are reflects of aluminosilicates and the presence of hematite and goethite. All clay materials consisted of a low amount of alkali and alkaline earth oxides: $K_2O$, $Na_2O$, MgO and CaO. The results of the physicochemical properties revealed that the clay materials were mostly constituted of a clay fraction (26–99%), followed by a silt fraction (1–70%) and sand fraction (0–4%). The textural classification corresponds to very heavy clays and silty clays. The studied samples were mostly plastic clays, with plasticity limit characteristics varying between 22.5% and 53% and a plasticity index ranging between 10.2% and 63.9%, which could be attributed to

the high organic matter content and the important proportion of clays. In the application, the studied clayey materials are promoted for the production of ceramics and terracotta building materials; in particular, pottery, bricks, tiles and stoneware. This study is therefore essential before any application of this type of clay in various industrial fields, specifically in fine ceramics.

**Author Contributions:** Conceptualization, N.N.G.F., M.M.B., T.N., B.E.C., M.J.S., E.J. and K.O.O.; methodology, K.O.O. and N.N.G.F.; formal analysis, K.O.O., N.N.G.F. and M.M.B.; investigations, K.O.O., N.N.G.F. and M.M.B.; data curation, T.N., B.E.C. and M.J.S.; resources, K.O.O., N.N.G.F. and M.M.B.; supervision, N.N.G.F. and E.J.; visualization, K.O.O., N.N.G.F. and T.N.; writing—original draft preparation, K.O.O., N.N.G.F. and M.M.B.; writing—review and editing, B.E.C., M.J.S. and N.N.G.F. All authors have read and agreed to the published version of the manuscript.

**Funding:** This research received no external funding.

**Institutional Review Board Statement:** Not applicable.

**Informed Consent Statement:** Not applicable.

**Data Availability Statement:** Data used in this article are available upon request to the corresponding author.

**Acknowledgments:** The authors are grateful to the reviewers for their scientific contributions and to those people who participated in different stages of the success of this manuscript.

**Conflicts of Interest:** We have no conflicts of interest. The authors agree with the information provided in the manuscript and the journal policies.

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
