# Peer review of "Mineralogical, Geochemical Characterization and Physicochemical Properties of Kaolinitic Clays of the Eastern Part of the Douala Sub-Basin, Cameroon, Central Africa"

_applsci, doi:10.3390/app12189143_

Round 1

Reviewer 1 Report

Dear Authors,

1. Please give more information in the abstract (number of samples, profile, brief method)

2. Figures 1 & 2: not complete 

3. Method: Provide a description of the triangular used in this study

4. Provide statistical analysis method & result. No statistics were done for the results.

5. Very minimal description of the profiles. Provide detailed description accordingly  

6. Missing important Physico-chemical parameters such as pH, CEC 

7.  Some statements in the Conclusion are not supported by the analysis undertaken

Author Response

Responses to the Reviewer 1 comments

Point 1. Please give more information in the abstract (number of samples, profile, brief method)

Response 1.

Three profiles from 3.9 to 7.4m thickness were studied on the field in order to determine their mineralogical (X-ray diffraction, infrared), chemical (X-ray fluorescence) and physicochemical (particle size, Atterberg limits, organic matter, cation exchange capacity and hydrogen potential) properties. Globally, ten (10) clay samples were analyzed to know the nature and properties of these clayey materials.

Point 2. Figures 1 & 2: not complete.

Response 2.

Figures 1 has been completed: Figure 1. Geological map of Douala modified [30]: (a) Location map of the Douala sub-basin in Cameroon; (b) Geological map of the Douala sub-basin including the study site; (c) Geological map of the study area

Figures 2 has been completed: Morphology of profiles of the clayey materials of the eastern part of the Douala sub-basin. a) Profile Y1 of Yansoki with analyzed samples Y1C1, Y1C2, Y1C3; b) Profile J1 of Japoma with analyzed samples J1C1, J1C2, J1C3; c) profile N1 of Ndogpassi with analyzed samples N1C1, N1C2, N1C3, N1C4.

Point 3. Method : Provide a description of the triangular used in this study.

Response 3.

The predominance of oxides was determined by using the triangular diagram SiO2-Al2O3-Fe2O3. The evaluation of the degree of chemical alteration of the studied materials was carried out by the ternary diagram of the evaluation of the chemical alteration index [34]. Weakly altered materials are with values between 50 and 60%, moderately altered materials 60 and 80%, highly altered materials > 80%. The certainty of the clayey materials in ceramics has been demonstrated by using the Fiori and Fabbri diagram [35].

The texture of the clay materials was determined using the Belgian textural classification diagram [36]. The porosity and permeability were determined by the McManus triangular diagram [37]. The prediction of the ceramic suitability of clay materials was determined using Winkler diagram [38].

The binary diagram of Holtz and Kovacs [40] was used to classify the studied clay materials according to their plasticity and the Bain diagram [41] was used as a support to show the suitability of the studied clays in ceramics using the data of the plasticity index of Atterberg: IP < 15% for bricks and IP > 15% for pottery.

Point 4. Provide statistical analysis method & result. No statistics were done for the results.

Response 4.

The samples were sent for analysis of major elements using the X-ray fluorescence spectroscopy method to the Cimencam laboratory, by the AFARGE group France figuil in the North Cameroon region. The results you see in Table 2 are such received. The statistical method you are talking about is not mentioned in the work.

Point 5. Very minimal description of the profiles. Provide detailed description accordingly.

Response 5.

Depending on the length of the manuscript, it would be better to limit the texts based on the essential information carried by the profiles which are the thicknesses, the color, the texture and the number of layers. All this information is continuous in Figure 2. Section 3.1. gives the description of the profiles.

Three holes (Y1, J1 and N1) measuring 8.7, 8.8 and 6.5 meters deep respectively were dug with a spade according to the availability of study sites and the positions at which some craftsmen exploit clay materials for their activities. Each profile was described and divided based on their texture and color (Munsell color chart) in different horizons or layers in the vertical sections. These profiles are essentially constituted of clayey horizons situated on the bottom of the profiles. All the profiles are surmounted by some sandy or conglomerated horizons may alternate with the rusty horizons and organo-mineral horizon. After describing the profiles, seventeen (17) samples of 5 kg each were taken manually with a shovel in the center of the different horizons encountered (Figure 2).

Point 6. Missing important Physico-chemical parameters such as pH, CEC

Response 6.

The Cation Exchange Capacity (CEC) was obtained by the Kjeldahl method [42], using ammonium acetate at pH 7 in three phases: saturation of the absorbent complex by NH4+ ions and extraction of exchangeable bases; washing the floor with alcohol to remove excess NH4+ ions; determination of NH4+ by Kjeldahl distillation after desorption from a KCL solution. pH measurement took place in two steps: firstly, measuring the active (or real) acidity (pH in water or pH-H2O) and secondly, measuring the potential acidity (pH-KCl). For pH-H2O, 10g of the specimen was mixed with 25ml of water and agited and there after left to stand for 24 hours before using a pH meter to measure the pH values. For the pH-KCl, 10g of the specimen was mixed with 25ml of a solution of KCl, agitated and left to stand before measuring the pH with a pH-meter. The pH-KCl is generally less than pH-H2O. The gap between these two pH values makes it possible to determine the reserve (or total) acidity of clay materials.

Point 7.  Some statements in the Conclusion are not supported by the analysis undertaken

Response 7.

In the eastern part of the Douala sub-basin, the exploratory field trip (Yansoki, Japoma and Ndogpassi) exhibits good indices of clay materials for ceramic with thickness of more than 7 meters over an area of the sub-basin. The geochemical, mineralogical and physicochemical characterizations were the investigated methods on clay materials. The morphological description of the raw clays deposit is made up of various clay facies (dark gray, light gray, purple, yellow and Multicolored). Mineralogically, the main clay minerals of the studied samples are constitued of kaolinite (48.3 - 69.2 wt.%) and quartz (20.5 - 41.2 wt.%) with lesser amounts of illite content (2.4 to 10.2 wt.%). Other minerals are also observed such as hematite, goethite, feldspar, gibbsite and micas, but of small to moderate amount. This mineralogical result suggested that monosiallitisation is the main crystallochemical process acting in the study area. Geochemical result showed that, SiO2 (22.21% - 58.03%) and Al2O3 (12.84% - 22.94%) are the main oxides with lesser to higher amounts of Fe2O3 content (1.07% - 17.92%). the significant amount of these oxides are reflects of aluminosilicates and the presence of hematite and goethite. All clays materials a low amount of alkali and alkaline earth oxides K2O, Na2O, MgO and CaO. The results of the physicochemical properties revealed that the clay materials are mostly constituted of clay fraction (26 - 99%), followed by silt fraction (1 - 70%) and sand fraction (0 - 4%). The textural classification corresponds to very heavy clays and silty clays. The studied samples are mostly plastic clays, with plasticity limit characteristics varying between 22.5% and 53% and plasticity index ranging between 10.2% and 63.9%, which could be attributed to the high organic matter content and fine particle size in the raw clay materials. In the application, the studied clay materials are suitable for the production of ceramics and terracotta building materials in particular, pottery, bricks, tiles and stoneware.  This study is therefore essential before any application of this type of clay in the various industrial fields specifically in fine ceramics.

Other responses to the notifications on the manuscript.

The notifications requested in the manuscript have been reviewed and corrected.

Reviewer 2 Report

An interesting piece of work. I believe it should be suitable for publication after the XRD and XRF analyses are better described and some changes to text, mostly typos and minor changes to the English language text.

I recommend that the following changes be made to the text:-

Page 1. 

1. Introduction.

Lines 2 and 6 start sentences with a capital letter, Clay not clay.

Line 7. Currently not Currenty.

Line 12. compositions not composition

Page 2.

Line 6. Change faïence (francais) to earthenware (english).

Line 7. Change interpreter to interpret.

Page 4. 

3.2. Analytical Techniques.

Unité de Recherche: Argiles, Géochimie et Environnements sédimentaires (AGEs) - should these people be included in the author list?

last line change x-ray diffraction to X-ray powder diffraction

Page 5.

Lines 3 and 4 show 2 to 45 degrees 2theta. However, Figure 4 shows XRD data collected out to 70 degrees 2theta, please give the correct 2theta range for data collections.

How long did it take to measure XRD patterns on each sample? How were samples mounted? Flat plate or capillary, was the sample spinning during data collection?

Line 5. placed not laced.

Lines 9-10. Qualitative and semi-quantitative estimates were referred to measurements of maximum intensity of X-ray models according

Does this mean qualitative and semi-quantitative estimates of phase proportions were determined from XRD data? Please rewrite this section and include more information.

Lines 15-16

Laboratory of Cimencam of Figuil in Cameroon, Lafarge group, France. should these people be included in the author list?

Line 17. X-ray fluorescence spectrometry

How were XRF analyses done? What sort of XRF spectrometer was used, what X-ray source was used? What was the elemental range that could be analysed. How were the samples prepared for XRF. Were samples analysed as received or prepared as pressed pellets or fused beads? Please give more information.

Last paragraph. Replace K2 with KHCO3. For H2 which acid, give this information.

Last but one line. distilled water for 24 hours - was this done at room temperature? Please indicate temperature.

Page 6

4. Results and Discussions

4.1. Mineralogy

line 4.

quartz, kaolinite, illite, hematite, muscovite, feldspar, goethite and gibbsite.

replace with 

quartz (SiO2), kaolinite (Al2(OH)4Si2O5), illite ((K,H3O(Al,Mg,Fe)2(Si,Al)4O10[(OH)2,(H2O)]), hematite (Fe2O3), muscovite (KAl2(AlSi3O10)(F,OH)2), feldspar (KAlSi3O8 – NaAlSi3O8 – CaAl2Si2O8), goethite (FeOOH) and gibbsite (Al(OH)3). Amend the stoichiometries where appropriate.

Line 5. semi-quantitative - were these determined by XRD?

Line 10. Replace lessly expressed with in lower proportions

Line 12. Replace present with shown to be present.

Line 12. Replace in with at

Lines 12-13. indicate low crystallinity - low crystallinity is indicated by broad peaks, not less intense peaks. Please amend.

Line 13. Replace his by its.

Line 15. Replace quantities give yellow with which at these proportions gives a yellow

Line 16. How much hematite is present?

Line 25. Replace cristallized with crystalline.

Line 25. What do you mean by average?

Lines 27-28. Replace These minerals are accompanied by quartz mainly with These minerals are mainly accompanied by quartz.

Line 30. Replace 'in the form of domes' with 'with broad peaks'

Line 34. Replace were with indicate the presence of 

Line 35. Replace minerals can be observed with mineral.

Line 37. Replace welldefined with well defined.

Page 7. Table 1 caption.

Semi-quantitative estimation of the mineralogical composition of clay materials of the eastern part of the Douala sub-basin.

Was this done by XRD? If so please indicate in the Table 1 caption.

Page 8. 

4.2. Geochemistry

Line 1 replace chemical analysis with XRF chemical analyses?

Page 9.

Line 9. K2O subscript 2

Line 10. Table 2 caption

Geochemical composition of clay materials of the eastern part of the Douala sub-basin (wt.%). 

Were these determined by XRF?

Page 14. Text in Figure 11. 

porosity not prosity. Remove acute accent from e in permeability

Page 16.

Organic matters

Line 2. presented replace with and these results are presented in Table 3.

Line 4. Replace size are fine with sizes are fine.

Line 4. block size - what do you mean by block size? Is this referring to particle size?

Page 17. Replace Conclusion with Conclusions, more than one conclusion presented!

Line 3 replace are constituted with consist of

Line 5 replace amount with amounts.

Line 7 replace and Fe2O3 is moderated content with with lesser amounts of Fe2O3.

Author Response

Responses to the Reviewer 2 comments

An interesting piece of work. I believe it should be suitable for publication after the XRD and XRF analyses are better described and some changes to text, mostly typos and minor changes to the English language text.

Response

The XRD and XRF analyses are better described and some changes to text, mostly typos and minor changes to the English language text according to your comments, you will have in the manuscript relevant modifications.

I recommend that the following changes be made to the text: Page 1. Introduction. Lignes 2 et 6. Clay

. Introduction.

Lines 2 and 6 start sentences with a capital letter, Clay not clay.

Response : Clay

Line 7. Currently not Currenty.

Response : This part has been removed

Line 12. compositions not composition

Response : compositions

Page 2.

Line 6. Change faïence (francais) to earthenware (english).

Response : earthenware

Line 7. Change interpreter to interpret.

Response : determine

Page 4. 

3.2. Analytical Techniques.

Unité de Recherche: Argiles, Géochimie et Environnements sédimentaires (AGEs) - should these people be included in the author list?

Response : The mineralogical analyses were carried out at the laboratory AGEs of the University of Liege, Belgium according to the methodology of Moore Duane and Reynolds Robert [32].

last line change x-ray diffraction to X-ray powder diffraction

Response : X-ray powder diffraction

Page 5.

Lines 3 and 4 show 2 to 45 degrees 2theta. However, Figure 4 shows XRD data collected out to 70 degrees 2theta, please give the correct 2theta range for data collections.

Response :

XRD data collected out at 2° to 70° 2theta.

Figure 4. XRD data collected out 2° to 70° 2theta

How long did it take to measure XRD patterns on each sample? How were samples mounted? Flat plate or capillary, was the sample spinning during data collection?

Response : The mineralogical analyses were carried out at the laboratory AGEs of the University of Liege, Belgium according to the methodology of Moore Duane and Reynolds Robert [32]. The mineralogy was determined by X-ray powder diffraction on bulk samples and on oriented clay samples at room temperature using a classical powder diffractometer Brüker Advance 8 brand type, equipped with Cu radiation (Kα ʎCu ¼ 1.54056 Å) at an intensity of 40 mA and a voltage of 40 kV. It is configured with a step size of 0.013° (2θ) for a measurement time of 30 s and data are collected over the interval of 2°–70° (2θ). Mineral phase identification was performed using X’Pert HighScore Plus software associated with PDF-2 2007 release software. Prior to analysis, the analyzed samples were ground and sieved to 80 μm. The < 2µm fraction was taken from the suspension after a settling time calculated according to Stocke’s law, placed on a glass slide and the XRD patterns were thus recorded between 2° and 30° 2θ using the same step size and time per step parameters. These oriented aggregates were subjected to three successive treatments: air drying, glycolation and heating at 500°C for 4h, in order to confirm the type of clay mineral phases.

Line 5. placed not laced.

Response : placed.

Lines 9-10. Qualitative and semi-quantitative estimates were referred to measurements of maximum intensity of X-ray models according

Response : Qualitative and semi-quantitative estimates (±5 -10%) were referred to measurements of maximum intensity of X-ray models according to Biscaye [33].

Does this mean qualitative and semi-quantitative estimates of phase proportions were determined from XRD data? Please rewrite this section and include more information.

Response : This part has been removed

Lines 15-16

Laboratory of Cimencam of Figuil in Cameroon, Lafarge group, France. should these people be included in the author list?

Line 17. X-ray fluorescence spectrometry

How were XRF analyses done? What sort of XRF spectrometer was used, what X-ray source was used? What was the elemental range that could be analysed. How were the samples prepared for XRF. Were samples analysed as received or prepared as pressed pellets or fused beads? Please give more information.

Response : Response : The geochemical characterization consisted of determining the major element concentrations using the X-ray fluorescence spectrosmeter (XRF). This analysis was performed on the raw materials using a Niton XL3t980 hXRF analyzer (X-ray tube: 50 kV; anode: silver; silicon detector: 8 mm). The analysis provides raw data in the form of a spectrum with the specific fluorescence energy (in keV) on the x-axis and the photon number (in CPS) on the y-axis. The instrument has been calibrated and configured in mining/mineral mode. The materials studied were crushed and sieved to 80μm and then analyzed to obtain the base oxides contained in the materials.

Last paragraph. Replace K2 with KHCO3. For H2 which acid, give this information.

Response : The organic carbon content is determined by the Walkley method, which is an oxidation with potassium bicarbonate (K2Cr2O7) in an acid medium (H2SO4).

Last but one line. distilled water for 24 hours - was this done at room temperature? Please indicate temperature.

Response : distilled water for 24 hours at room temperature.

Page 6. 4. Results and Discussions

4.1. Mineralogy

line 4.

quartz, kaolinite, illite, hematite, muscovite, feldspar, goethite and gibbsite. replace with quartz (SiO2), kaolinite (Al2(OH)4Si2O5), illite ((K,H3O(Al,Mg,Fe)2(Si,Al)4O10[(OH)2,(H2O)]), hematite (Fe2O3), muscovite (KAl2(AlSi3O10)(F,OH)2), feldspar (KAlSi3O8 – NaAlSi3O8 – CaAl2Si2O8), goethite (FeOOH) and gibbsite (Al(OH)3). Amend the stoichiometries where appropriate.

Response : quartz (SiO2), kaolinite (Al2(OH)4Si2O5), illite ((K,H3O(Al,Mg,Fe)2(Si,Al)4O10[(OH)2,(H2O)]), hematite (Fe2O3), muscovite (KAl2(AlSi3O10)(F,OH)2), feldspar (KAlSi3O8 – NaAlSi3O8 – CaAl2Si2O8), goethite (FeOOH) and gibbsite (Al(OH)3).

Line 5. semi-quantitative - were these determined by XRD?

Response : the semi-quantitative mineralogical composition of the mineral phases of the clay materials based on peak intensities of XRD patterns.

Line 10. Replace lessly expressed with in lower proportions

Response : in lower proportions

Line 12. Replace present with shown to be present.

Response : shown to be present.

Line 12. Replace in with at

Response : at

Lines 12-13. indicate low crystallinity - low crystallinity is indicated by broad peaks, not less intense peaks. Please amend.

Response : This part has been removed

Line 13. Replace his by its.

Response : its

Line 15. Replace quantities give yellow with which at these proportions gives a yellow

Response : which at these proportions gives a yellow

Line 16. How much hematite is present?

Response : hematite (0 – 3.8 wt.%)

Line 25. Replace cristallized with crystalline.

Response : crystalline.

Line 25. What do you mean by average?

Response : low peaks

Lines 27-28. Replace These minerals are accompanied by quartz mainly with These minerals are mainly accompanied by quartz.

Response : These minerals are mainly accompanied by quartz

Line 30. Replace 'in the form of domes' with 'with broad peaks'

Response : 'with broad peaks'

Line 34. Replace were with indicate the presence of 

Response : indicate the presence of

Line 35. Replace minerals can be observed with mineral.

Response : mineral.

Line 37. Replace welldefined with well defined.

Response : well defined.

Page 7. Table 1 caption.

Semi-quantitative estimation of the mineralogical composition of clay materials of the eastern part of the Douala sub-basin.

Response : based on peak intensities of XRD patterns

Was this done by XRD? If so please indicate in the Table 1 caption.

Response : based on peak intensities of XRD patterns

Page 8. 

4.2. Geochemistry

Line 1 replace chemical analysis with XRF chemical analyses?

Response : XRF chemical analyses

Page 9.

Line 9. K2O subscript 2

Response : K2O

Line 10. Table 2 caption

Response : Response : determined by XRF

Geochemical composition of clay materials of the eastern part of the Douala sub-basin (wt.%). 

Response :

Were these determined by XRF?

Page 14. Text in Figure 11. 

porosity not prosity. Remove acute accent from e in permeability

Response : Porosity, moderately, permeability

Page 16.

Organic matters

Line 2. presented replace with and these results are presented in Table 3.

Response : and these results are presented in Table 3.

Line 4. Replace size are fine with sizes are fine.

Response : sizes are fine.

Line 4. block size - what do you mean by block size? Is this referring to particle size?

Response : However, the organic matter has no effect on the final ceramic product when particles sizes are fine, but when the particle size are large, they show wide pores and carbon marks on product after firing [64].

Page 17. Replace Conclusion with Conclusions, more than one conclusion presented!

Response : Conclusions

Line 3 replace are constituted with consist of

Response : are consisted of kaolinite

Line 5 replace amount with amounts.

Response : amounts.

Line 7 replace and Fe2O3 is moderated content with with lesser amounts of Fe2O3

Response : with lesser to higher amounts of Fe2O3 content (1.07% - 17.92%).

Reviewer 3 Report

applsci-1885807-peer-review-v1

Physicochemical and Mineralogical Characterization of the Clay Materials of the Eastern Part of the Douala Sub-Basin (Cameroon, Central Africa)

Comments and Suggestions for Authors

The paper entitled " Physicochemical and Mineralogical Characterization of the Clay Materials of the Eastern Part of the Douala Sub-Basin (Cameroon, Central Africa)" is based on the X-ray diffraction, infrared, and X-ray fluorescence. The article can be considered only if it is revised.

1. The abstract must be written completely again. I found nothing attractive in the abstract. The abbreviations are not defined. 

2. Do not use brackets in the title.

3. The authors discussed several times what they are going to propose but did not discuss the research gap or what are the sole reasons for this research. 

4. Introduction section should be completely updated by adding motivation, organization and novelty of the work.

5. A comparative study if included would have an impact.

6. The authors wrote the conclusion in a rush. Please describe your achievements in detail. 

7. The quality of the figures is too low. I did not understand the results in Figure 5. Please explain and them and redraw it.

8. The length of this article is too long, it is recommended to cut down to the standard number of pages.

9. Avoid repetitions. I can see several repetitions at different places in this paper. Thorough proofreading is required.

**

Author Response

Responses to the Reviewer 3 comments

  1. The abstract must be written completely again. I found nothing attractive in the abstract. The abbreviations are not defined.

Response 1 :

The clay materials of the eastern part of the Douala sub-basin (Cameroon) were studied to determine their mineralogical composition and physicochemical properties to boost their potential suitability as materials for traditional ceramics and eventually modern ceramics. These clayey materials are not widely used locally as building materials and little data is available on these materials in the field of ceramics and they are relatively unknown. Three profiles from 3.9 to 7.4m thickness were studied on the field in order to determine their mineralogical (X-ray diffraction, infrared), chemical (X-ray fluorescence) and physicochemical (particle size, Atterberg limits, organic matter, cation exchange capacity and hydrogen potential) properties. Globally, ten (10) clay samples were analyzed to know the nature and properties of these clayey materials. Mineralogically, kaolinite (48.3 - 69.2 wt.%) and quartz (20.5 - 41.2 wt.%) are the most abundant minerals in these raw clay materials. They are associated to a very small or moderate quantity of illite, hematite, goethite, feldspar, gibbsite and micas. Geochemically, the clayey materials have high silica (SiO2, 22.21% - 58.03%) and alumina (Al2O3, 12.84% - 22.94%) contents with significant amount of iron oxides (Fe2O3, 1.07% - 17.92%). Other oxides (K2O, MgO, TiO2, Na2O, MnO, CaO and P2O5) are in relatively lower proportion. High level of alumina content explains the kaolinitic nature of these clayey materials. The results of physicochemical properties indicate that granulometric analysis of clayey materials show the following distribution: clay (26–99%), followed by silt (1–70%) and sand (0–4%). This corresponds to silty-clay soils according to the Belgian textural classification diagram, with high plastic index (63.9%) characteristics. The studied clay materials are the good candidates for the production of ceramics and terracotta building. This study is therefore important before any application of this type of clay in the various industrial fields.

  1. Do not use brackets in the title.

Response 2 : The brackets have been removed in the title.

  1. The authors discussed several times what they are going to propose but did not discuss the research gap or what are the sole reasons for this research.

Response 3 : In Cameroon’s coastal basins, the studies have been carried out for the physicochemical and mineralogical characterization of clay materials in order to understand their origin and paleoenvironment [23, 24] and to determine their properties for various applications [25, 26, 27]. In the present context, the nature and physicochemical and mineralogical properties of the clay materials in the Douala sub-basin are not very well know and little works have been done on the ceramic aspect.

The aim of this study is to carry out a mineralogical and physicochemical characterization of the clay materials at the eastern part of the Douala sub-basin (Cameroon, Central Africa) in order to determine their nature and identify their technological classification and suitability for various ceramic applications.

  1. Introduction section should be completely updated by adding motivation, organization and novelty of the work.

Response 4 : The traditional use of clayey materials exists to ancient time in Africa (Egypt), Asia (China), America (Mexico), Europe (Roman), etc [1]. Clay is a material which has particle size less than 2μm and belongs to the family of minerals with similar chemical compositions and common crystal structural characteristics [2]. Raw clay materials are mineral resources with major importance in industries because of their numerous uses [3, 4]. Presently, more than a third of the world’s population uses clayey products, due to their quality, weather resistance, plasticity and malleability. The behavior of clay materials is related to their mineralogy and chemical composition, associated with certain geotechnical characteristics (particle size, plasticity, etc.). Their chemical compositions vary depending on both the physical and chemical changes in the environment where clay deposits are found [5]. The use of clay raw materials has grown in several and varied applications (paints, plastics, cosmetics, pharmaceuticals and production of ceramic materials of wide distribution) [6, 7, 8]. Indeed, the fields of application of clay minerals are more and more numerous because they are used in several industrial sectors, the most developed of which are the ceramic industry for the manufacture of porcelain, earthenware, the agricultural industry to dilute pesticides and the pharmaceutical industry for the manufacture of medication [6, 7, 8]. Studies have been carried out on the applications of ceramic clays in several areas around the world [9, 10, 11, 12]. However, several researchers have been interested in the ceramic applications of clay materials in Africa [13, 14, 15]. Several works have been focused on clay raw materials in Cameroon. Clay materials have been identified in several geological environments in Cameroon [16, 17, 18, 19, 20]. Other works have been carried out on physicochemical and mineralogical characteristics of the catalytic properties and thermal behavior of some clay materials [21, 22]. In Cameroon’s coastal basins, the studies have been carried out for the physicochemical and mineralogical characterization of clay materials in order to understand their origin and paleoenvironment [23, 24] and to determine their properties for various applications [25, 26, 27]. In the present context, the nature and physicochemical and mineralogical properties of the clay materials in the Douala sub-basin are not very well know and little works have been done on the ceramic aspect.

The aim of this study is to carry out a mineralogical and physicochemical characterization of the clay materials at the eastern part of the Douala sub-basin (Cameroon, Central Africa) in order to determine their nature and identify their technological classification and suitability for various ceramic applications.

  1. A comparative study if included would have an impact.

Response 5 :

Section 4.1.

 The mineralogical composition of the samples in this study is similar to that of the Douala clay materials [25, 26]. However, the results obtained were compared with other clay materials which were formed under the same conditions notably the raw clay materials of Ketou Benin [47], Lokoundje Cameroon [48] and Tamazert, Hadj Ali and Chekfa from Algeria [49]. The same mineralogical characteristics of the mineral assemblage were observed except that the rutile concentration was found in Lokoundje Kribi Cameroon [48. The results from XRD analysis can be confirmed by FTIR spectra (Figure 6) indicating the presence of kaolinite, quartz, illite and oxyhydroxide mineral. The absorption peak found between 3700 cm−1 and 3600 cm−1 (3692.60 cm−1, 3615.23 cm−1) in all samples attests the presence of kaolinite [48, 28]. The lack of a well defined peak at 3664 cm−1 may be due to the existence of disordered kaolinite [50]. The peaks of quartz found between 1200 cm−1 and 900 cm−1 (1115.98 cm−1, 1006.31cm−1, 904 cm−1) for all samples because of the stretching vibration of Si-O that confirms the presence of this mineral and which corroborates with the results from XRD analysis [51]. However, the absorption at 3364.33 cm−1 and 740 cm−1 indicates the presence of illite [52]. These characteristics confirmed the origin of sediments of the studied samples [50].

  1. The authors wrote the conclusion in a rush. Please describe your achievements in detail.

Response 6 : In the eastern part of the Douala sub-basin, the exploratory field trip (Yansoki, Japoma and Ndogpassi) exhibits good indices of clay materials for ceramic with thickness of more than 7 meters over an area of the sub-basin. The geochemical, mineralogical and physicochemical characterizations were the investigated methods on clay materials. The morphological description of the raw clays deposit is made up of various clay facies (dark gray, light gray, purple, yellow and Multicolored). Mineralogically, the main clay minerals of the studied samples are consisted of kaolinite (48.3 - 69.2 wt.%) and quartz (20.5 - 41.2 wt.%) with lesser amounts of illite content (2.4 to 10.2 wt.%). Other minerals are also observed such as hematite, goethite, feldspar, gibbsite and micas, but of small to moderate amount. This mineralogical result suggested that monosiallitisation is the main crystallochemical process acting in the study area. Geochemical result showed that, SiO2 (22.21% - 58.03%) and Al2O3 (12.84% - 22.94%) are the main oxides with lesser to higher amounts of Fe2O3 content (1.07% - 17.92%). the significant amount of these oxides are reflects of aluminosilicates and the presence of hematite and goethite. All clays materials a low amount of alkali and alkaline earth oxides K2O, Na2O, MgO and CaO. The results of the physicochemical properties revealed that the clay materials are mostly constituted of clay fraction (26 - 99%), followed by silt fraction (1 - 70%) and sand fraction (0 - 4%). The textural classification corresponds to very heavy clays and silty clays. The studied samples are mostly plastic clays, with plasticity limit characteristics varying between 22.5% and 53% and plasticity index ranging between 10.2% and 63.9%, which could be attributed to the high organic matter content and fine particle size in the raw clay materials. In the application, the studied clay materials are suitable for the production of ceramics and terracotta building materials in particular, pottery, bricks, tiles and stoneware. This study is therefore essential before any application of this type of clay in the various industrial fields specifically in fine ceramics.

  1. The quality of the figures is too low. I did not understand the results in Figure 5. Please explain and them and redraw it.

Response 7 : Figure 5 has been redrawed and explanations have been provided in its legend.

In section 4.1.

Figure 5 presents the XRD patterns of the < 2µm fraction of analyzed samples.

In Figure 5, kaolinite is characterized by diffraction peaks between 7.11 and 7.18 Å in the normal state (N) and after treatment with ethylene glycol (EG), then disappears after heating to 500°C (CH). This clay mineral has a sharp diffraction peak, indicating that it is crystalline. However, illite is indicated by low peaks between 9.56 and 10.3 Å at normal (N) and do not change after treatment with ethylene glycol (EG) and heating to 500°C (CH). These minerals are mainly accompanied by quartz. According to Hinckley [46], kaolinite and quartz peaks on the diffractograms are well expressed indicating that the mineral have been well crystallized. However, other similar peaks are identified in raw clay materials with broad peaks or weakly expressed that are characteristic of minerals with a lower degree of crystallinity.

  1. The length of this article is too long, it is recommended to cut down to the standard number of pages.

Response 8 : All repeated passages in the manuscript have been deletedwhich has reduced a good deal of text in this paper.

  1. Avoid repetitions. I can see several repetitions at different places in this paper. Thorough proofreading is required.

Response 9 : A thorough proofreading was done and rehearsals were removed.

Round 2

Reviewer 3 Report

No comment